# Observational Study on a Large Italian Population with Lipedema: Biochemical and Hormonal Profile, Anatomical and Clinical Evaluation, Self-Reported History

**DOI:** 10.3390/ijms25031599

**Published:** 2024-01-27

**Authors:** Laura Patton, Lorenzo Ricolfi, Micaela Bortolon, Guido Gabriele, Pierluigi Zolesio, Erika Cione, Roberto Cannataro

**Affiliations:** 1Endocrinology and Lymphology Clinic, 38096 Vallelaghi, TN, Italy; lauraptt77@gmail.com (L.P.); lorenzoricolfi@hotmail.it (L.R.); 2Rehabilitation Unit and Lymphology Clinic, Institute San Gregorio, 31049 Valdobbiadene, TV, Italy; micaelabortolon@gmail.com; 3Department of Medical Biotechnology, University of Siena, 53100 Siena, SI, Italy; guido.gabriele@unisi.it; 4Lymphology Clinic, 09129 Cagliari, CA, Italy; zolesiop@tiscali.it; 5Department of Pharmacy, Health and Nutritional Sciences, University of Calabria, 87036 Rende, CS, Italy; erika.cione@unical.it; 6Galascreen Laboratories, University of Calabria, 87036 Rende, CS, Italy; 7Research Division, Dynamical Business & Science Society—DBSS International SAS, Bogotá 110311, Colombia

**Keywords:** lipedema, adipose tissue, pain, hormonal impairment, obesity, inflammation, ultrasound

## Abstract

We analyzed the medical condition of 360 women affected by lipedema of the lower limbs in stages 1, 2, and 3. The data were assessed for the whole population and compared between different clinical stages, distinguishing between obese and non-obese patients. The most frequent clinical signs were pain when pinching the skin, subcutaneous nodules, and patellar fat pads. The most frequently painful site of the lower limbs was the medial lower third of the thigh. The pain score obtained on lower limb points increased progressively with the clinical stage. In all points evaluated, the thickness of the subcutaneous tissue increased with the clinical stage. Analyzing the data on the lower medial third of the leg and considering only patients with type 3 lipedema, the difference between stages was statistically significant after correction for age and BMI. We found higher levels of C-reactive protein at more severe clinical stages, and the difference was significant after correction for age and BMI between the stages. Overall, the prevalence of alterations of glucose metabolism was 34%, with a progressive increase in prevalence with the clinical stage. The most frequent comorbidities were vitamin D insufficiency, chronic venous disease, allergies, dyslipidemia, headache, and depression of mood. Interestingly, in comparison with the general population, we found higher prevalence of chronic autoimmune thyroiditis and polycystic ovary syndrome. Finally, the clinical stage and the involvement of the upper limbs or obesity suggest a worse clinical, anthropometric, and endocrine–metabolic profile.

## 1. Introduction

Lipedema is a chronic pathology, defined for the first time by Allen and Hines in 1940, but still not well known and difficult to diagnose. The disease typically affects the female sex, and it can potentially involve any part of the body; it is often confused with lymphoedema, lipodystrophy, or obesity [1,2].

The true prevalence of lipedema is unknown, although it is thought to be a common disease. The estimated prevalence ranges from 6.5% in children in the United States and 6−8% in women in Germany to 15−19% in vascular disease clinics [3]. It is an inflammatory and fibrotic disease of the loose connective tissue (of which the adipose tissue is a part) that generally spares the trunk and does not affect the hands and feet. From this preferential anatomical distribution derives a typical disproportion between the extremities and the trunk due to a localized and symmetrical increase in the subcutaneous adipose tissue at the level of the lower and/or upper limbs [4].

Lipedematous adipose tissue differs from healthy adipose tissue by the presence of inflammation, which is associated with spontaneous or provoked fibrosis and pain and which, in some cases, can lead to alterations in sensitivity in the affected area. The fibrotic component seems responsible for the incongruous volumetric reduction of the affected limbs despite weight loss obtained with nutritional measures, physical activity, or bariatric surgery [3].

Overweight, obesity, a sedentary lifestyle, lymphatic and venous insufficiency, hormonal treatments, pregnancy, and menopause are considered aggravating or trigger factors rather than causes of the disease. Some endocrine dysfunctions, such as hypothyroidism and hypercortisolism, could be further detrimental factors, particularly if associated with overweight, obesity, mobility limitation, and other manifestations such as myxedema [2]. There is no scientific evidence that lipedema has a progressive course or a constant rate of worsening over time [5].

Lipedema is divided into five types based on the anatomical location where lipedema tissue is present [3]. Two distinct forms of lipedema are also recognized: the “columnar” form, where the cylindrical conformation of the limbs is typical, and the “lobar” form, where there are large lobes of tissue protruding from the side view of the limbs [6].

Instead, the staging of lipedema refers to the degree of clinical severity and mainly includes three distinct clinical pictures [3] (Figure 1).

Stage 1: Skin has a smooth texture with a subdermal pebble-like feel due to underlying loose connective tissue fibrosis.Stage 2: Women have more lipedema tissue than women at stage 1 and skin dimpling due to the progressed fibrotic changes and excess tissue. Palpable nodules may be more numerous and more considerable.Stage 3: Skin features increased lipedema tissue that is more fibrotic in texture, with numerous large subdermal nodules and overhanging lobules of tissue.

If lymphological alterations arise in a particularly advanced stage 3, some authors define this clinical picture with the term lipo-lymphedema or stage 4. At the same time, when alterations of the lymphatic system arise and remain parallel to the lipedema, and therefore can also be present in the initial stages, some authors define the clinical picture with the term lympho-lipedema, a clinical condition in which the two pathologies coexist in the same patient [7].

The diagnosis of lipedema is clinical and is carried out through the evaluation of diagnostic criteria helpful in framing the characteristics of the pathological tissue. The disease is characterized by a volumetric increase in the affected body area bilaterally and symmetrically and by the presence of nodules of fibrous tissue in the subcutaneous tissue of variable dimensions, more prone to the onset of spontaneous bruises and ecchymoses or due to minimal trauma. Spontaneous pain, pain provoked by touch and/or a sense of heaviness, and transient edema (aggravated by orthostasis) in the limbs may be present, as well as livedo reticularis and joint hypermobility. At the same time, the Stemmer’s sign is negative [3,4].

A differential diagnosis should be made with edematous fibrosclerotic panniculopathy (EFP), obesity, lymphedema, some rare diseases of the adipose tissue (Dercum’s disease, Madelung’s disease), polycystic ovary syndrome, Cushing’s syndrome [2], growth hormone deficiency, and lipodystrophies associated with lipohypertrophy [8]. Furthermore, it cannot be excluded that some of these diseases may coexist or that the presence of endocrine diseases that cause or contribute to obesity may at the same time also favor the onset of lipedema in predisposed subjects, as suggested by the genetic mutations found in patients affected by lipedema in genes already known associated with syndromes or forms of genetic obesity, rare adipose tissue diseases or connective tissue diseases [9,10].

The standard conservative therapy for lipedema includes specific physiotherapy and nutritional treatments, compression garments, and specific motor physical activity plans [3]. Compression therapy has always been and remains an essential element of the “best practice” in the treatment of patients with lipedema [5]. Until 15–20 years ago, confused with obesity (which is sometimes present as comorbidity), it was treated with nutritional programs aimed at weight loss, but with poor or unsatisfactory results; the ketogenic approach (a nutritional program that involves the intake of minimal quantities of carbohydrates) is now being successfully used [11,12]; the success is probably due to an intrinsic action of the ketogenic diet in modulating inflammation [13], but, obviously, also to the total absence of glycemic peaks; the use of supplements is often considered, but the rationale for use is often anecdotal, so few are considered truly useful [14].

This study aimed to analyze the medical history and the clinical, ultrasound, and laboratory characteristics of a large number of Italian women affected by lipedema. No previous study has reported data on Italian women affected by lipedema.

## 2. Results and Discussion

### 2.1. General Description, Phenotype, and Staging

We analyzed data on 360 women affected by lipedema of the lower limbs. The age was between 12 and 76 years (mean age 40.4 ± 11.8 years), 78.3% of whom were well into childbearing age.

The stage distribution was as follows: stage 1—39.7% (143/360), stage 2—40.0% (144/360), stage 3—20.3% (73/360).

Type 3 of lipedema was the phenotype with the highest prevalence in 89.7% of cases (323/360). In the other cases, 8.3% (30/360) had type 2 lipedema, 1.1% (4/360) had type 5, and 0.8% (3/360) had type 1 lipedema.

In all cases analyzed, the lipedema was of the columnar type.

In our population, 54.7% of all patients (197/360) had also upper extremity involvement. The prevalence of upper extremity involvement in association with the lower extremities increased from stage 1 to stage 3 (*p* < 0.001): in stage 1, the upper limbs were involved in 35.7% of cases; in stages 2 and 3, the upper limbs were involved in 58.3% and 84.9%, respectively.

The mean age of the patients also increased with the severity of the clinical stage (*p* = 0.001) but was not significantly different between stages 2 and 3. Moreover, the mean age of patients with upper extremity involvement was higher than in those without (*p* = 0.003). At the time of the first visit, 32 patients had already undergone liposuction of the lower limbs.

### 2.2. Onset and Evolution of the Disease

In 57.8% of cases, the onset of the disease coincided with the menarche (202/360), while, in 22.8% of cases, it occurred before the menarche (81/360), in 16.1% of cases after the menarche (53/360), and in 3.3% coinciding with the menopause (12/360). In cases of onset before puberty (81/360), overweight was already present in almost 60% of cases. The mean age of onset was 13.4 ± 9.49 years, and we did not find a difference between the stages in patients with or without upper limb involvement (Table 1).

The mean disease duration was 27.01 ± 13.1 years, and it increased with the severity of the stages (*p* < 0.001).

Analyzing the evolution of the disease over time, the main factors that had a negative impact on the evolution of the disease were, in order of frequency, weight gain (any cause) in 53.9% (194/360), pregnancy and or breastfeeding in 53.04% (96 cases out of 181 patients who had a full-term pregnancy), and the use of hormonal contraceptives in 44.4% (102 of 262 who used hormonal contraceptives).

Although the disease can be reported in some cases as stable over time, most patients report the worsening of the disease, either gradually over the years or with phases of drastic worsening following the intake of hormonal contraceptives, pregnancy, or increased weight. With the discontinuation of hormonal contraceptives, in general, the symptoms, such as the sense of swelling, tension, and pain in the limbs, are reduced. However, the regression of increased adipose tissue in the limbs often no longer happens.

In women in a menopausal state (84 patients), later lipedema worsening at menopause was observed in 67.9% of cases (57/84). In nearly all cases of clinical worsening with menopause, concomitant weight gain was reported. In cases with onset before menarche (78 patients), pubertal development worsened the disease in 41% of cases. In contrast, it did not seem to have had a pejorative effect in the rest of the cases.

A study on 209 women affected by lipedema reported an average age of higher onset (16 ± 9 years of age); however, the data obtained from a questionnaire administered to the patients [15] may not be accurate. In our experience, patients often, when asked “When did the lipedema arise?”, report the age at which a deterioration occurred or the subjective symptoms arose, rather than the onset of adipose tissue hypertrophy in the affected sites.

### 2.3. Family History

A family history of lipedema was investigated in first-, second-, and third-degree relatives: a positive family history was found in 81.2% (289/356) of cases. Of these cases, in 40.2%, the familiarity was maternal, in 32.6% paternal, and in 4.8% both maternal and paternal; in 3.7% (13/356), the disease was present only in daughters or sisters. A family history of lipedema was absent in 18.8% of cases (67/356). There were no differences in family history between the clinical stages or between patients with and without upper limb involvement.

In the literature, positive familiarity relating to first- and second-degree relatives is reported in 15–64%, depending on the study, and is generally considered present in over 60% of cases [8,16].

### 2.4. Anthropometry and Endocrinological Clinical Evaluation

Table 1 shows the data relating to anthropometry and other endocrinological clinical aspects. The prevalence of obesity, defined with BMI (BMI ≥30 kg/h^2^), was 38.9% (140/360), with the following distribution: obesity class 1 (BMI ≥ 30 and < 35) 20.3%, obesity class 2 (BMI ≥ 35 and < 40) 10.8%, and obesity class 3 (BMI ≥ 40) 7.8%. As expected, the prevalence of obesity increased with the severity of the stage: 6.3% in stage 1, 44.4% in stage 2, and 91.8% in stage 3 (*p* < 0.000). The prevalence of obesity was higher in patients with upper limb involvement (56.9% and 17.2%, respectively, *p* < 0.000).

Given the characteristic disproportion between the lower limbs and trunk in patients with lipedema, other adipose tissue distribution indices were evaluated, indicative of a possible association of the disease with abdominal obesity and alterations of metabolism and cardiovascular risk factors. The presence of an increase in waist circumference was evaluated: a waist circumference > 80 cm, a cut-off considered diagnostic for visceral obesity according to the International Diabetes Federation (IDF) and the European Association for the Study of Obesity (EASO), was found in over 80% of patients (84.9%), while a waist circumference > 88 cm, a value considered equivalent to the presence of BMI > 30 kg/m^2^ by the European Society of Endocrinology (ESE), was found in 63.9%. Two other indices that can be useful in phenotyping the distribution of adipose tissue is the ratio between the waist circumference and hip circumference (WHR = waist circumference /hip circumference) and the ratio of the waist circumference to height (WHtR = waist circumference/height). The cut-off used for WHR correlated to greater cardiometabolic risk, 0.85 [17], and for WHtR it was 0.5 [18]. In our population, a pathological WHR (WHR ≥ 0.85) was found in 29.3% of cases, while a pathological WHtR (WHtR > 0.5) was found in over 82% of the population (80.4%). The prevalence of patients with an increased waist circumference greater than 80 cm and greater than 88 cm, as well as with an increased WHtR, significantly increased with the severity of the clinical stage (*p* < 0.001 for all three parameters) and in patients with affected upper limbs compared to patients without (*p* < 0.001 for all three parameters). On the other hand, there were no differences between stages regarding the prevalence of a pathological WHR. At the same time, the prevalence was significantly higher if lipedema was also present in the upper limbs (39.8% vs. 17.2%, *p* < 0.001). The increase in waist circumference in these patients could also be due, at least in part, to the apposition of suprafascial subcutaneous lipedema tissue at the abdominal level, especially under the umbilicus; however, these data will be investigated with targeted methods also for possible implications on the stratification of the cardio-metabolic risk of these patients. Acanthosis nigricans was detected in 32.3% of the patients. In all cases, acanthosis nigricans affected the elbow; in more than half of the cases, it was isolated to this site. The prevalence of acanthosis nigricans increased with the clinical stage (18.2%, 33.3%, and 58.3%, respectively, *p* < 0.001) and was higher in the case of upper extremity involvement (40.8% vs. 22.1%, respectively, *p* < 0.001). Another interesting finding was the presence of a dorsal hump in 17.4%, with an increase in the clinical stage (5.4%, 21%, and 33.8%, respectively, *p* < 0.000) and in patients with upper limb involvement (29.5% vs. 2.1%, *p* < 0.000). Hirsutism, defined by the presence of a Ferriman-Gallwey score > 8 [19], was found in 19 patients (5.30% of cases). The score was equal to 0 in 63.8% of the cases (with glabrous skin detected in the areas affected by lipedema) and less than 3 points in over 80% of the patients. There are no data in the literature regarding the prevalence of acanthosis nigricans, a dorsal hump, and/or hirsutism in the population affected by lipedema. The prevalence of hirsutism was lower than that reported in the studies present in the literature (4.2% vs. 13%) [20,21]. These data are interesting as they could be related to the hormonal balance, particularly the estrogen/androgen ratio in these patients.

### 2.5. Clinical Lymphological Aspects

Table 2 shows the data relating to lymphological clinical aspects. In 92.5% of cases, at least one of the specific symptoms of lipedema was reported by the patient: spontaneous pain in 70.0% of cases, and sensations of heaviness, swelling, and fatigue in the absence of pain in the remaining cases. In most cases, the symptoms worsened during the day, following physical activity, especially if of high intensity, but also in situations of prolonged time spent in a sitting position or prolonged time in a standing position, during the luteal phase of the menstrual cycle, and, in general, with the warm season. Only in 10% of cases were the symptoms unaffected by the described factors. Meanwhile, some of the patients reported an improvement in symptoms with physical activity. Bruising due to minimal trauma or spontaneously was reported in 80.8% of cases.

Table 2 shows the prevalence of symptoms and clinical signs in the entire population (ALL), in the population divided into the three clinical stages (STAGE 1, STAGE 2, STAGE 3), and in the two groups of patients without involvement of the upper limbs (WITHOUT UP) and with upper limb involvement (WITH UP). We found no differences between the groups for the subjective symptomatology described before.

In our population, the three most frequently encountered clinical signs were pain on pinching the skin of the affected limbs, evoked in 99.4% of cases (358/360); the presence of subcutaneous nodules on palpation, present in 98.9% of cases (351/348); suprapatellar fat pads in 93.8% (331/353); a dimple in the medial retromalleolar region in 86.7% (294/339); and perimalleolar fat pads, present in 87.2%. All these clinical signs, except for pain on pinching, discussed below, were more frequent with more severe clinical stages.

We also found the same trend for the prevalence of telangiectasia, ankle cuff signs, pretibial fovea, and increased back foot folds (Table 2).

The prevalence of the presence of pain on pinching the skin in the lower extremities did not differ between stages or in patients with/without upper extremity involvement, except for the prevalence of pain in the medial upper third of the thigh and lower abdomen between stages (*p* = 0.029) and in the lateral upper third of the thigh and lower abdomen, where it was more frequent in those with affected upper limbs. The pain from pinching the skin, systematically evaluated at the lower and upper limbs, abdomen, and back, was absent in all sites only in two specific cases: a patient who had recently undergone liposuction of the limbs and a young patient (12 years old) evaluated for a family history of lipedema, diagnosed in the mother. A third patient did not have pain in the lower limbs but it was present in the arms, with a diagnosis of lipedema type 1 and 4.

The most frequently painful site of the lower limbs was the medial lower third of the thigh, where pain was evoked in 98% of cases (99.3% in patients with type 3 lipedema and not considering patients who previously underwent liposuction). However, considering the locations evaluated for the lower limbs, pain was evoked in at least 95% of all the other sites that were evaluated, except for the trochanteric region and abdomen, where it was present in 90.6% of cases, and the abdomen (present in 28.6% of cases).

The pain was quantified, as previously described, using a Verbal Rating Scale with values from 0 to 4 (0 = no pain, 4 = maximum pain), applied to each specific site evaluated, and we calculated three different scores: a lower limb pain score, an upper limb pain score, and a total pain score. The mean of all three pain scores increased with the severity of the clinical stage (*p* < 0.001) (Figure 2). The mean of the lower limb score was as follows: stage 1, 2.57 ± 0.808; stage 2, 2.96 ± 0.696; stage 3, 3.17 ± 0.676 (*p* < 0.001). In this case, the difference was significant when comparing stage 1 with stages 2 and 3; it was not significant when comparing stage 2 with stage 3. However, considering the total pain score, the difference was significant between all stages (*p* < 0.001). It was maintained even after correction for BMI and age.

As expected, the upper limb pain score was higher in patients with upper limb lipedema (upper pain score 0.234 ± 0.450 vs. 1.97 ± 0.968, respectively, *p*< and 3.06 ± 0.757). However, if lipedema was present in the upper limbs, the pain score in the lower limbs was also higher (lower pain score 2.59 ± 0.714 vs. 3.06 ± 0.757, respectively, *p* < 0.001), as well as the total pain score (total pain score 1.88 ± 0.543 vs. 2.73 ± 0.712, respectively, *p* < 0.001). The presence of lipedema in the lower limbs would, therefore, seem to indicate a more severe clinical condition in the lower limbs.

Considering the points evaluated on the lower limbs, the site with the highest mean score was the medial upper third of the leg ), with a mean of 3.65 ± 0.671 points.

Clinical signs of chronic venous disease were also evaluated: overall, considering the presence of clinical signs (telangiectasia, venous varicose, dyschromia, phlebostatic crown) and the instrumental findings by color Doppler ultrasound, the prevalence of chronic venous disease was 71.9%. The prevalence of chronic venous disease increased with the severity of the clinical stage (62.2%, 75.0%, and 84.9%, respectively, *p* = 0.001). The presence of joint hypermobility was evaluated with the Beighton score, using a score ≥ 5 as a cut-off for diagnosis [22]: joint hypermobility was detected in 31.59% (73/232) of our population.

The comparison with the data available in the literature regarding the detection of other signs and symptoms appears more complex due to the lack of data and homogeneity in the evaluation of clinical signs and due to heterogeneous populations, which could be more comparable. A study carried out on 148 patients is available to compare some of the clinical signs that we evaluated in our population [23]: the only comparable data are related to the prevalence of the ankle cuff, found in 75%, higher prevalence than in our population. Telangiectasias is reported in over 50% of the population examined [8]. Chronic venous disease is considered the most common vascular problem found in patients with lipedema, even if data in the literature are scarce, with prevalence ranging from 25 to 50% depending on the study [3]. The prevalence of joint hypermobility in a study of 160 women with lipedema was greater than 50% [24]. There are no objective data on the detection of pain in the subcutaneous adipose tissue fold that can be compared, which is why we introduced the detection of pain on pinching the skin with a quantitative score. The validity of this method needs to be tested on a population of women affected by lipedema that is more homogeneous in terms of stage, type, and clinical history. However, from a clinical point of view, it could be helpful to define the disease’s clinical severity better and evaluate its progress over time.

### 2.6. Ultrasound Evaluation

The thickness of the subcutaneous tissue was measured in nine sites of the lower limb. Considering all the patients evaluated, the thickness of the skin increased significantly with the clinical stage at the level of all points, except for the area at the trochanter level. The thickness of the adipose tissue increased progressively with the clinical stage, with a significant difference in all nine points evaluated (*p* < 0.001 for all points).

Even in the comparison between patients with and without the involvement of the upper limbs, the thickness of the tissue increased significantly at the level of all points. At the same time, this was only observed at the level of the medial upper third of the leg and the inguinal level due to the thickness of the skin (*p* = 0.006 and *p* = 0.001, respectively).

As expected, comparing type 2 with type 3 (groups of patients with more significant numbers), the thickness of the adipose tissue was significantly different at the level of the lower third and upper medial third of the leg (*p* < 0.001 and *p* = 0.004, respectively) and the level of the upper lateral third of the leg. At the same time, there were no differences in the other points.

Table 3 shows the results of skin and adipose tissue measurements in a subpopulation of 102 patients whom we selected as suffering from type 3 lipedema, who had not previously undergone liposuction, who had not followed diets, or who had body weight reductions >10% of weight and with blood tests available. Data are shown for the entire subpopulation and the population divided by clinical stage. Except for the measurement of the skin at the trochanter level and the upper third of the lateral leg, the tendency towards an increase in its thickness with the severity of the clinical stage was confirmed. The progressive increase in the thickness of the adipose tissue was significant at all levels of the lower limb evaluated. It was always significant in the comparison between individual groups, except for the comparison between stages 2 and 3 at the level of the upper medial third of the leg and between stages 1 and 2 at the upper and lower lateral third of the leg.

Considering the measurement of adipose tissue referring to the lower medial third of the leg in this subpopulation, it was possible to analyze the correlation with other parameters concerning the clinical history and the blood tests. This subpopulation of 102 patients, aged between 13 and 67 years (average 40.4 ± 11.2 years) with a BMI between 20 and 56 (average 27.6 ± 13.1), the following distribution by stage was found: stage 1, 37.3% (38/102); stage 2, 39.2% (40/102); stage 3, 23.5% (24/102).

The thickness of the adipose tissue at this level varied from 7.1 mm to 53 mm (mean 22.1 ± 7.6 mm) and increased progressively with the severity of the stage (*p* < 0.001). The thickness of the adipose tissue at this level correlated with the thicknesses measured in all other points of the lower limb (*p* < 0.001 for all points). The thickness was not statistically correlated with age, age of onset, disease duration, and years of treatment with estrogen-progestins. There were no differences in thickness between those who had a family history of lipedema or obesity, those who had used or not used estrogen-progestins, and those who had spontaneous pain or not. The thickness, as expected, correlated positively with all anthropometric parameters, i.e., BMI (*p* < 0.001), waist circumference and hip circumference (*p* < 0.001), and WHrT (*p* < 0.001), while it did not correlate with WHR. The thickness correlated positively with the total pain score (*p* = 0.003).

Regarding laboratory tests, the thickness did not correlate with ACTH levels, cortisol, or thyroid function. We instead found a positive correlation with CRP levels (*p* < 0.001), IGF-1 levels (*p* = 0.017), fasting insulin (*p* = 0.029), HOMA-IR (P0.041), and blood sugar at 120 min of the OGTT (*p* < 0.001). There was also a negative correlation with HDL levels (*p* < 0.003). All correlations listed were no longer significant after correction for BMI. The negative correlation that emerged between the thickness of the adipose tissue in the lower medial third of the leg and the peak blood sugar level at 30 min of the OGTT could be interesting. However, it needs to be explored further to understand its meaning. This negative correlation remained significant even after correction for BMI. Among these 102 patients, we had 82 patients of fertile age that we used for the evaluation of the correlation with sexual steroids (estrogens and androgens) without hormonal therapy: the thickness did not correlate with the levels of gonadotropins, estradiol, or progesterone, the main androgens.

There is little research using ultrasound on lipedema, and it is not easy to compare our results. In a previous study, Amato et al. established clinical applicability criteria with simple and reproducible ultrasound cut-off values for the diagnosis of lipedema in the lower limbs. They suggested a cut-off of 11.7 mm for pre-tibial region thickness measurements, with better accuracy, followed by a cut-off of 17.9 mm for the thigh and a cut-off of 8.4 mm for the lateral leg thickness for the diagnosis of lipedema. The optimal cut-off value for the supramalleolar thickness was 7.1 mm, but the accuracy of the diagnosis at this level was lower than at the other points. The supramalleolar thickness showed statistically significant differences between individuals with and without lipedema, but when reanalyzed using stratification by BMI, the supramalleolar thickness was not statistically significant in the normal weight subgroup. Nevertheless, in our population, in no case was the thickness measurement less than 7.1 mm [25].

Marshall et al. suggested the classification of lipedema based on the thickness of the subcutaneous tissue measured at this point: from 12 to 15 mm thickness, it was compatible with lipohyperplasia or mild lipedema; from 15 to 20 mm, it was compatible with moderate lipedema; a thickness greater than 20 mm was compatible with an indisputable diagnosis of lipedema; and one greater than 30 mm with severe lipedema [26]. Although the cut-off used to define a severe clinical condition of lipedema was similar to what was found in stage 3 patients in our population, it is impossible to make a direct comparison due to the different clinical classifications. Some other studies have been performed to identify ultrasound features, such as the thickness and echogenicity of skin and subcutaneous tissue, for the differential diagnosis between lipedema and lymphedema [27].

In our work, ultrasound was used, in support of the evaluation, as a first instrumental approach to exclude the diagnosis of lymphedema by evaluating the dermal thicknesses, the presence and the aspect of inguinal lymph nodes, and the thickness and appearance of the subcutaneous tissue.

In our opinion, the ultrasound measurement of the thickness of the subcutaneous superficial adipose tissue at the level of the lower medial third of the leg could be a simple and quick measurement to perform, which, as was found, correlates with the clinical stage of the disease in type 3 lipedema. The ultrasound study of adipose tissue is a simple and repeatable method that helps to define the disease’s clinical severity and could be used for the monitoring of the evolution of the disease and the response to treatment over time. However, these data must be re-evaluated in more homogeneous and comparable populations.

### 2.7. Blood Tests

Table 4 shows the results of the main blood tests performed in the entire population and the groups as described before. Table 5 shows the hormonal test results for the part of the population that included only patients of childbearing age, due to the known effect of menopause on the hormones evaluated.

#### 2.7.1. General Blood Test and Inflammation Markers

Regarding biochemical examinations, no relevant alterations were found in the complete blood count (CBC) examination, liver and kidney functionality, electrolytes, total protein, and electrophoresis.

An increase in C-reactive protein levels (CRP), related to a cut-off of 5 mg/L, was detected in 15 patients (15/378 patients). Of these ten patients, 2 had stage 1, 2 had stage 2, and 11 patients had stage 3. When comparing clinical stages, a significant difference was found in CRP levels, which increased with the clinical stage (1.38 ± 1.29 mg/L, 1.94 ± 1.69 mg/L, 4.93 ± 4.35 mg/L, respectively, *p* < 0.001). The difference in CRP levels between stages was not significant when comparing stage 1 and stage 2. The CRP levels were also significantly higher in patients with upper limb lipedema (3.16 ± 3.27 mg/L in patients with upper limb lipedema, 1.422 ± 1.50 mg/L in patients without, *p* < 0.001). The difference in CRP levels remained significant between clinical stages after the correction of the analysis for age and BMI between the stages. At the same time, it lost its significance in the comparison between patients with and without upper limb involvement. In our population of women with lipedema, the CRP levels were higher if obesity was present (1.41 ± 1.49 mg/L in non-obese subjects, 3.91 ± 3.53 mg/L in obese subjects, *p* < 0.001). The increase in mean CRP levels increased with the BMI range.

In a part of the population, we also evaluated the levels of the C3 and C4 complement fractions, and it is interesting to note that also, in this case, we detected a significant increase in levels with the clinical stage and a higher level of both proteins in patients with involvement of the superior limbs. The difference in the levels of these two proteins was no longer significant after correction for BMI, except the levels of the C4 complement fraction, which remained significant in the comparison between patients with and without involvement of the upper limbs, being higher in those with lipedema also in the upper limbs (Table 3).

Data relating to indices of acute and low-grade inflammation in patients with lipedema are few, and it is difficult to compare results in patients with different clinical stages or phenotypes. A recent study reported an increase in circulating inflammation markers and oxidative stress markers in women with lipedema compared to age- and BMI-matched healthy controls [28]. The increasing trend in CRP levels and C4 fractions that we found, which remained significant even after correction for BMI, could be related to the extent of the inflamed pathological adipose tissue and the impact that the disease could have on systemic metabolic and inflammatory conditions.

#### 2.7.2. Lipid Metabolism

An increase in the levels of total cholesterol (>200 mg/dL), triglycerides (>150 mg/dL), and LDL (>130 mg/dL) was found in 36.9% (76/206), 4.5% (9/200), and 29.7% (55/185), respectively, while reduced levels of HDL (<50 mg/dL) were found in 12.4% of cases or 26/210. Overall, dyslipidemia was present in 47.3% of the population, with hypercholesterolemia predominating, without differences between the groups. We found a significant difference in the levels of HDL between the clinical stages (*p* = 0.007) and between patients with lipedema in the upper limbs and patients without affected upper limbs (*p* = 00.1). Patients with lipedema in the upper limbs also had higher levels of triglycerides (*p* = 0.006). These differences were no longer significant after correcting the analysis for age and BMI for both comparisons.

As regards the prevalence of dyslipidemia, comparing the available data relating to the Italian adult female population aged between 35 and 74 years [29], the prevalence of hypercholesterolemia, hypertriglyceridemia, and LDL hyperlipidemia and reduced HDL levels (68.9%, 16.2%, 68.4%, 15.5%, respectively) would appear to be lower in a patient with lipedema. However, the data must be confirmed with a study between patients with lipedema and comparable unaffected controls, at least comparable in age and BMI. The data confirm what has already been reported in the literature, i.e., low prevalence of dyslipidemia despite a high BMI [30].

#### 2.7.3. Glucose Metabolism

Fasting hyperglycemia (glycemia > 100 mg/dL, according to the criteria indicated in the document Italian Standards for the treatment of diabetes mellitus 2018) [31] was found in 16.9% of cases (7/169).

There was no report of an increase in fasting glucose and/or glycated Hb levels compatible with a diagnosis of diabetes mellitus [31,32], nor was the presence of diabetes reported in the anamnesis.

A 75 g oral glucose tolerance test was also performed in 170 patients with the dosage of glycemia and insulin in basal conditions. After 120 min, we identified one case of diabetes mellitus (blood sugar > 200 mg/dL at 120 min) and seven cases of impaired glucose tolerance (blood sugar > 140 mg/dL at 120 min) [31,32].

A condition of fasting insulin resistance, diagnosed by the HOMA-IR index (HOMA-IR = (glucose mg/dL × insulin mIU/L)/22.5) [33] using 2.29 as a cut-off [34], was present in 30.7% of cases (62/140). The presence of insulin resistance was also assessed using a surrogate marker of insulin resistance, Stumvoll 0–120, which was calculated from the blood glucose and insulin values detected with OGTT in basal conditions and 120 min after the glucose load considering the cut-off of 0.080 indicative of insulin resistance [35,36]: we noted a condition of insulin resistance in 7% of patients (10/141). In all these cases, the diagnosis of insulin resistance was already present based on the baseline values calculated with the HOMA-IR. Considering the OGTT, we detected a significant difference when comparing the Stumvoll index mean 0–120 (*p* = 0.047) and blood glucose levels at 120 min after the glucose load (*p* = 0.003), both worsening with the clinical stage. However, the difference was not maintained after correction for BMI. Moreover, no differences emerged between the clinical stages in the AUC of glycemia and insulin or the other points of the OGTT for glycemia and insulin (Table 4). As regards the load curve, the only difference that emerged in the comparison between patients with and without involvement of the lower limbs concerned the average peak insulin level after 60 min from the glucose load (*p* = 0.037), which, even in this case, was not significant after the correction of the analysis for BMI. However, it should be kept in mind that these data should be re-evaluated, and the effect of current diets (particularly if ketogenic) or significant weight changes that occurred before the blood tests should also be considered.

Overall, the prevalence of glucose alterations considering the cases of fasting hyperglycemia, fasting insulin resistance, and impaired glucose tolerance was 39.2% (79/231). Considering the overall prevalence of glucose metabolism alterations, the difference was significant between the stages (*p* = 0.001), and the prevalence increased with the severity of the clinical stage: 22.5% in stage 1, 32.5% in stage 2, and 52.9% in stage 3. Moreover, comparing the patients with and without involvement of the upper limbs, a significant difference emerged (*p* = 0.012): 26.7% in the case of involvement of the lower limbs only and 41.7% in the case of involvement of the upper limbs.

The analysis was repeated in the population excluding patients who had undergone low-calorie or low-carb diets in the last six months (154 patients): the prevalence was 33.1% (51/154). In this population, the mean levels of HOMA-IR, fasting insulin, and fasting glucose progressively increased with the severity of the clinical stage (*p* < 0.001, *p* < 0.001, and *p* = 0.014, respectively). However, the difference was not statistically significant if corrected for BMI. We also found higher HOMA-IR and fasting insulin levels in the patients with upper limb involvement (*p* = 0.002 and *p* = 0.001, respectively), without significant differences in fasting glucose levels. Moreover, in this case, the difference was not significant if corrected for BMI. Finally, the prevalence of glucose metabolism disorders was higher in obese subjects than in non-obese subjects (50.0% vs. 23.5%, *p* = 0.001). It increased with the BMI range: 24.4% in normal-weight patients, 22.8% in overweight, 38.7% in obesity class 1, 56.3% in obesity class 2, and 77.8% in obesity class 3.

The prevalence of diabetes in the Italian female population reported in the literature is 4.2% (ages 18–69) [37], with prevalence that increases with age. Therefore, our population would appear to be lower, since we found only one case of diabetes. Moreover, the prevalence of hyperglycemia, evaluated in this case, considering a cut-off for glycemia of ≥110 mg/dL as indicated by the World Health Organization [38] for comparison with the data available on the Italian female adult population, was lower (3.9% in our population versus 6.1% as reported) [37]. In a previous study, the authors reported higher fasting insulin levels in women with lipedema compared to age- and BMI-matched healthy controls. However, no other indices of glucose metabolism have been studied [28]. It Is difficult to compare the data with those of people not affected by lipedema; however, the overall prevalence of glucose metabolism alterations in the entire population observed by us remains high, affecting 34% of the population, and it should not be overlooked, especially in more severe clinical cases and in cases of obesity. The data will have to be confirmed in other studies and compared with a larger and more homogeneous population of women not affected by lipedema, but what we have already found, in our opinion, is relevant from a clinical point of view, as the presence of insulin resistance could in itself favor the onset or worsening of the disease or interfere with the effect of treatment in many respects, or it could be a consequence of the inflammatory state of lipedema.

#### 2.7.4. Insulin-like Growth Factor-1 (IGF-1)

We found a difference in insulin-like growth factor-1 (IGF-1) levels between the three clinical stages: the mean decreased with the severity of the clinical stage (*p* < 0.001). The difference was significant when comparing stage 1 with stages 2 and 3 (*p* = 0.006 and *p* < 0.001, respectively) but not when comparing stage 2 with stage 3. The same statical result was found even when comparing patients with and without upper limb involvement (*p* = 0.001), with lower mean values in patients with upper limb involvement. The difference between the clinical stages and between patients with and without upper limb involvement was no longer significant when correcting for age and BMI. Moreover, we found that IGF-1 levels are markedly lower if obesity was present (IGF-1 132 ± 55.2 μg/L in obese subjects and 178 ± 78.0 μg/L in non-obese subjects, *p* < 0.001). However, the reduction in IGF-1 levels did not progressively decrease with the BMI range. We found a negative correlation between IGF-1 levels and BMI, while the correlation was neither present with fasting insulin levels nor HOMA-IR.

In obese people, it is known that there is a reduction in the spontaneous and stimulated secretion of GH and also the concomitant presence of an acceleration in GH clearance. In obese people, reduced GH levels are correlated with a worse cardiovascular risk profile and obesity-related comorbidities, also due to the effect on abdominal and visceral obesity [39]. The effects of GH are predominantly mediated by IGF-1. However, GH affects the metabolism; in particular, GH potently stimulates lipolysis, and IGF-1 does not mediate this effect. It is important to note that the reduction in GH levels is considered a secondary effect of obesity and that, despite what one might expect, the total levels of IGF-1 in obesity are not reduced; the opposite is true. This paradox can be explained by the elevated insulin levels in the obese, which stimulates IGF-1 synthesis and suppresses its transport protein (IGF-binding protein 1).

Consequently, the increased free and active IGF-1, induced and sustained by insulin, could explain the suppression of GH secretion by negative feedback [40]. There are no studies that have investigated the axis regarding GH/IGF-1 in patients with lipedema; however, an in vitro study performed on adipose stem cells obtained from lipoaspirate demonstrated a difference in the expression of IGF-1 during the proliferative activity. The IGF-1 levels were initially higher in stem cell cultures from patients with lipedema in comparison with control stem cells, but a reverse trend was found after the stimulation of adipogenesis. IGF-1 is considered an essential regulator of adipose cell differentiation, particularly in the terminal phase of the process. The role of IGF-1 in the pathogenesis of lipedema has been postulated [41]. In our opinion, these data emerged because the GH/IGF-1 axis could be involved in the pathogenesis of lipedema and because it could also help in the differential diagnosis of simple obesity.

#### 2.7.5. Vitamin D

The presence of insufficient vitamin D levels was calculated considering the cases detected with blood tests (25-OH-Vitamin D < 30 ng/mL) [41] and the history of hypovitaminosis while taking vitamin supplements. The insufficiency of vitamin D was found in 84.6% of our population, with no differences between clinical stages or between patients with and without upper limb involvement. This deficiency was the most prevalent finding among the comorbidities detected in this population (Table 5).

However, it was not surprising, considering that vitamin D deficiency is frequent in Italy, reaching values of 65% in young subjects and up to 86% in the female population over 70 years old [42], and it is more significant in obese subjects. In fact, in addition to factors relating to lifestyle, pathophysiological conditions linked to excess fat mass also contribute to determining the deficit. Numerous epidemiological studies, including the National Health and Nutrition Examination Survey (NHANES) III and the Framingham study and clinical trials, have found increased prevalence of hypovitaminosis D with increasing BMI. Therefore, the prevalence of hypovitaminosis in women affected by lipedema was higher and more significant than those reported in the literature, and this could be due to a storage or metabolic defect in the adipose tissue affected by lipedema. In our opinion, this is a problem that should not be underestimated and should be treated as a possible concomitant and worsening factor of the symptoms of lipedema, being able to favor, for example, the pain in the limbs and the fatigue often described by patients.

### 2.8. Thyroid Function

Hypothyroidism was found in 22.5% of patients (81/279), considering the patients already in treatment with a history of hypothyroidism and the new cases found with blood tests.

The prevalence of hypothyroidism increased with the clinical stage (16.1% in stage 1, 22.9% in stage 2, 34.2% in stage 3, *p* = 0.001), while there was no difference between those with and without upper limb involvement. We also detected a history of hypothyroidism in 24 other cases, one third of which were affected by chronic autoimmune thyroiditis.

If we considered all the cases (previous and confirmed hypothyroidism), the prevalence of hypothyroidism was about 30% (105/360). The leading cause of hypothyroidism was autoimmune thyroiditis, as found in about 56% of cases (*p* < 0.001).

An unexpected finding was the high prevalence of autoimmune thyroiditis (presence of anti-thyroglobulin and/or anti-peroxidase antibodies), which we found in 35.5% of our population (77/237), with no differences between clinical stages.

Given that hypothyroidism is often more frequent in obese subjects than in normal-weight subjects, the prevalence of hypothyroidism was analyzed as a function of BMI: the prevalence of hypothyroidism was higher in obese patients than in normal-weight and overweight patients (29.3% versus 18.2% respectively, *p* = 0.010). However, analyzing the data in patients with different classes of obesity, the prevalence did not increase progressively with the degree of the obesity: 18.9% in normal weight, 17.5% in overweight, 31.5% in class 1 obesity, 25.6% in class 2 obesity, and 28.6% in class 3 obesity. The higher prevalence of hypothyroidism between obese and non-obese subjects could also be explained by the possible presence of hypothyroidism secondary to obesity in this group of patients.

The prevalence of hypothyroidism in the general population reported in the literature varies from 5 to 10% with the gender, age, and population studied. It increases with age and is higher in females [43]. The prevalence of hypothyroidism in obese subjects is higher than in the non-obese population; it ranges from 1.7 to 43.7%. An average of 14% was estimated for both overt hypothyroidism and subclinical hypothyroidism, and BMI was not positively correlated with the prevalence of overt hypothyroidism in a meta-regression [44].

Regarding thyroiditis, in a study on a healthy population, anti-TG antibodies were detected in 10.4% of cases and anti-TPO antibodies in 11.3%. The prevalence of autoimmune thyroiditis is 5–10 times greater in women than in men [45].

In women affected by lipedema, the higher prevalence of hypothyroidism has already been reported in the literature compared to the general population, with values of 27–36% depending on the study [3]. The data were also confirmed in our population, although with lower prevalence than other studies, even previously carried out by us on fewer patients. It is higher than in the general population, and it progressively increases with the severity of the clinical stage and in obese subjects.

The prevalence of autoimmune thyroiditis in women with lipedema has never been evaluated, and it was higher in our study than reported in the literature for the general population (32.6% vs. 10–11%). This finding could be relevant, considering that an association between autoimmune diseases and other adipose tissue diseases has been described, as in the case of acquired generalized lipodystrophy [46]. Lipodystrophy syndromes are rare and heterogeneous disorders characterized by the complete or partial deficiency of adipose tissue. Based on the evidence of a positive association between acquired lipodystrophies and autoimmune disorders, including immune-mediated alterations in the adipose tissue of patients affected, a reaction against white adipose tissue antigens is postulated [47]. Another important aspect could be the link between immune and inflammatory processes and adipose tissue.

In contrast, adipose tissue plays a critical role as an endocrine organ that produces several active peptides, including leptin and adiponectin, and numerous cytokines. For example, leptin’s serum levels and gene expression in adipocytes strongly correlate with the proportion of body fat stores, and it acts as pro-inflammatory adipokine inducing T helper 1 cells, possibly contributing to the development and progression of autoimmune responses. Adiponectin also acts as an anti-inflammatory factor, especially concerning atherosclerosis, but, in some chronic inflammatory/autoimmune diseases, adiponectin may have pro-inflammatory effects, and its production correlates with inflammatory markers and disease activity [48]. This aspect needs to be explored further with other studies.

### 2.9. Ovarian and Adrenal Hormone Profile

As previously described, the hormonal assays were performed in the follicular phase in all fertile patients and at least two months after the discontinuation of hormone therapy, while they were performed randomly in the case of amenorrhea or menopause.

Considering the dosages available in the whole population, no relevant prolactin, cortisol, and ACTH alterations emerged (Table 3).

No significant alterations in the levels of gonadotropins, estradiol, and progesterone emerged. The results were compatible with the cycle phase or the menopausal state. Table 6 shows the hormonal profile in women with lipedema (282 cases), excluding menopausal women. We evaluated the levels of del 17-OH-progesterone, androstenedione, and DHEA-S. The assay of total testosterone and SHBG was performed, calculating the free androgen index (FAI = T (ng/dL) × 3.47/SHBG (nmol/L) × 100) [49]. Considering a cut-off of 0.68 ng/mL for total testosterone and of 4.97 for FAI [50], we found increased levels in 3.73% (5/134) and 3.09% (3/97), respectively. An increase in androstenedione levels (with a cut-off of 350 ng/dL) was found in 4.76% (5/105, three isolated cases, two associated with an increase in total testosterone levels), an increase in DHEA-S in about 9% of cases (11/126, in a single case associated with elevated levels of testosterone and androstenedione, in the other cases isolated). In no case was an increase in 17-OH-progesterone levels detected (cut off of 200 pg/mL). The overall prevalence of hyperandrogenemia was 12.7% (17/134).

Analyzing the difference in the hormonal profile in the various clinical stages, we found a difference in the levels of FAI and 17-OH-progesterone (see Table 6). The difference was no longer significant after the correction of the analysis for BMI and age for FAI. At the same time, it was still significant between stages for 17-OH-progesterone levels, but there was no progressive trend in the levels of 17-OH-progesterone with the stages.

In comparing patients with and without involvement of the upper limbs, we found a significant difference in the levels of androstenedione, FAI, and 17-OH-progesterone. The difference was no longer significant after the correction of the analysis for BMI and age for FAI for all of these hormones.

It is difficult to determine whether these data are relevant, and they need to be confirmed by excluding the interference due to the use of a different laboratory. This aspect is particularly relevant if we study androgen levels in women, because the dosage was not administered with the method considered the gold standard (liquid chromatography and tandem mass spectrometry). A previous study found no significant differences in the circulating levels of sex hormones between lipedema patients and age- and BMI-matched control women [28]. However, the subjects recruited in the study were few. Considering the levels of total testosterone, FAI, androstenedione, and DHEA-S, the prevalence of hyperandrogenemia in our population was comparable to that estimated in the general population (12.7% versus 11%, respectively) [21]. One of our goals is to investigate this aspect further, as an imbalance between androgen and estrogen levels could underlie the pathogenesis or progression of the disease during life, and this is supported by the finding of lipedema in male subjects only in conditions of hypogonadism [4,51].

### 2.10. Other Aspects and Comorbidities

#### 2.10.1. Menstrual Cycle Alterations and Polycystic Ovary Syndrome

The presence of menstrual irregularities, specifically oligomenorrhea (cycle longer than 35 days) and/or polymenorrhea (cycle < 25 days), was reported in 32.5% of patients (117/360) (Table 5). The most prevalent alteration was oligomenorrhea, responsible for 75% of cases of menstrual irregularities and overall present in 24% of the population. We also report five cases of premature menopause.

The finding of polycystic ovaries in previous gynecological evaluations was reported in 19.2% of cases. Polycystic ovary syndrome (PCOS) was diagnosed in 17.1% of the population (54/315). The diagnosis was made according to the Rotterdam criteria [52,53] in the presence of at least two of the following parameters: clinical (hirsutism, acne, alopecia) and/or biochemical hyperandrogenism (biochemical hyperandrogenism refers to elevated total testosterone or FAI), chronic oligo-anovulation, and micro-polycystic ovary, after the exclusion of secondary causes.

The most prevalent phenotype of PCOS was type D, characterized by the presence of oligo-anovulation and PCO, responsible for 63.0% of PCOS cases (34/54) in our population. Phenotype A, characterized by the presence of hyperandrogenism (clinical or biochemical), oligo-anovulation, and PCO, was responsible for 16.7% (9/54) of cases, and phenotype B, characterized by the presence of hyperandrogenism and oligo-anovulation, for 14.8% (8/54). Finally, phenotype C, characterized by the presence of hyperandrogenism and PCO, was responsible for 6% of cases (3/54).

Compared to what was reported in the literature [21], the higher prevalence of oligo-anovulation was found in this population than in the general population (32.5% versus 15%, respectively), the lower prevalence of PCOS (19.2% versus 28%, respectively), and, as already reported, the lower prevalence of hirsutism (5.3% versus 13%, respectively). The prevalence of PCOS was overall increased in patients with lipedema in comparison with the general population (17.1% versus 10%, respectively), but with higher prevalence of a non-typical phenotype or one not associated with high androgen levels. This finding is relevant because the normoandrogenic PCOS phenotype is considered a phenotype with a lower metabolic risk profile (such as in terms of insulin resistance and diabetes) [54]. In a previous study on 209 women with lipedema [15], polycystic ovary syndrome was detected in 5.7% of cases, although it was not differentiated by phenotype. The data, however, were collected through a questionnaire, and this could explain the difference, in part, compared to what was detected in our population.

#### 2.10.2. Pregnancy and Other Gynecological Problems

Fertility problems were reported in 9.02% of cases (all causes of female infertility). Eclampsia and gestational hypertension have been reported in 11% of women with full-term pregnancies (19/172); in four of these patients, the problem occurred in two successive pregnancies. Gestational diabetes has been reported in 6.4% (11/172) of these women. (Table 5). These data are comparable with data reported on the European population for eclampsia and gestational hypertension (10% of all pregnancies) [55] and gestational diabetes (most often reported as 2–6% of pregnancies) [56].

An aspect that perhaps should be carefully evaluated, but certainly more in depth with other studies, could be that relating to spontaneous abortion, which, in our population, was reported in approximately 30% of patients. A review of the current evidence found that the pooled risk of miscarriage was 15.3% of all recognized pregnancies; the risk is lowest in women aged 20–29 years at 12%, increasing steeply to 65% in women aged 45 years and older [57]. It is difficult to determine whether these data could be relevant or not; they should also be re-evaluated considering the age of the woman when she had the abortion and the gestational age of the fetus, as with the other risk factors. However, conducting this analysis with the data at our disposal is impossible.

Endometriosis in our population was reported in 4.2% of cases (15/345), comparable with the prevalence described for the female general population (2–10%) [58]. Other gynecological problems are reported in Table 5; three cases of ovarian teratoma and one case of breast malignancy have also been reported.

There have been no other cases of malignant tumors in our population.

#### 2.10.3. Metabolic Syndrome

The diagnosis of metabolic syndrome was made according to the criteria of the International Diabetes Federation [59,60]: the presence of central obesity with waist circumference > 80 cm in association with two other criteria among arterial hypertension (systolic blood pressure ≥ 130 mmHg and/or diastolic blood pressure ≥ 85 mmHg), hypertriglyceridemia (triglycerides ≥ 150 mg/dL), low HDL cholesterol levels (HDL cholesterol < 50 mg/dL), or current treatment for dyslipidemia. The prevalence of metabolic syndrome was 6.64% (16/241).

The prevalence in the general female population has been estimated at 21% [37]; therefore, it would seem lower in women with lipedema despite the high prevalence of an increased waist circumference and obesity. The prevalence of metabolic syndrome increased with the severity of the clinical stage and in the case of the involvement of the upper limbs (*p* = 0.003 and *p* = 0.002, respectively).

#### 2.10.4. Other Comorbidities

Among the other relevant clinical problems with prevalence at least greater than 40%, we found allergies (both pharmacological and of other types), which affected more than 50% of our population, headache, mood disorders (reported by patients, no diagnostic tests performed), and bowel disorders (irregular bowel movements, constipation, irritable bowel syndrome, and/or other intestinal diseases) (see Table 5).

The prevalence of mood disorders increased with the severity of the clinical stage (39.7%, 42.8, 61.2%, respectively, *p* = 0.012) and in the case of the involvement of the upper limbs (57.6% vs. 35.2%, *p* = 0.001).

Other relevant problems were gastric disorders in 32.9% of cases (gastroesophageal reflux, dyspepsia, post-prandial swelling, chronic gastritis, and other gastric diseases) and urinary incontinence in 25.8%, with higher prevalence in the more severe clinical stages and in patients with upper limb involvement.

A history of previous fractures was reported in 29.0% of cases. A history of joint dislocations or sprains was reported in 20.9% of cases, but almost 33% of patients had joint hypermobility.

Only a few studies are available in the literature to compare with the data found in our population. A study of 209 patients [15], carried out through a questionnaire on a population of women affected by lipedema evaluated for liposuction, revealed the prevalence of allergies to be 34.4%, depression to be 23%, migraine to be 23%, and intestinal disorders to be 12.9%. Given the method of data collection, the real presence or prevalence of the pathologies could be determined in a more reliable way. Another interesting study is available in the literature on a large population of 245 women affected by lipedema, in which the prevalence of various comorbidities was evaluated through questionnaires. However, it is not easy to compare due to the different populations’ greater ages and the higher percentage of clinical stage 3 compared to our population [61].

### 2.11. Obesity and Lipedema

Within our population, we evaluated the differences between women with lipedema with obesity (BMI > = 30) and those without obesity (BMI < 30), also considering the impact that BMI was shown to have on the results described. In our population, obese patients had a higher mean age (*p* < 0.001) and a longer mean duration of disease (*p* < 0.001).

There was no significant difference in the age of onset of lipedema.

There was a difference in the distribution of the clinical stages (*p* < 0.001): in obese patients, the stage with the most significant prevalence was stage 3, found in 47.9%, while the prevalence of stage 1 was 6.4% and of stage 2 was 45.7%. In non-obese patients, the most prevalent stage was stage 1, found in 60.9% of cases, while that of stage 2 was 36.4% and that of stage 3 was only 2.7%.

The frequency of upper limb involvement was also higher in obese patients (80% vs. 38.6%, *p* < 0.001). No differences emerged in the family history of lipedema, while obese patients more often had a family history of obesity than non-obese patients (*p* < 0.001).

From a clinical point of view, as expected, obese patients had a greater waist and hip circumference and had reached a greater maximum weight during their lifetime (*p* < 0.001); they also showed acanthosis nigricans and dorsal humps more frequently, while there were no differences for hirsutism, acne, and irregularities of the menstrual cycle.

From a lymphological point of view, no differences emerged in the presence of less spontaneous pain or a tendency towards bruising (persistent symptoms across the entire population). However, we found more frequently subcutaneous nodules (*p* < 0.001), skin laxity (*p* < 0.001), ankle cuff signs (*p* < 0.010), medial retromalleolar dimples (*p* = 0.005), suprapatellar fat pads (*p* = 0.028), perimalleolar fat pads (*p* = 0.048), fovea (*p* = 0.001), and telangiectasias (*p* = 0.005). We found no differences in the prevalence of joint hypermobility.

However, the presence of pain caused by the fold of the skin on the upper limbs, back, and abdomen was more frequent (*p* < 0.001), while, at the level of the lower limb, the presence of pain was seen only at the level of the upper medial third of the thigh. The average pain and total pain scores in the lower limb and upper limb were significantly higher in obese patients (*p* < 0.001 for all three scores).

As regards blood tests, in patients with obesity, we found higher levels of fasting blood glucose (*p* = 0.003) and after 120 min on the OGTT (*p* = 0.003) and of fasting insulin (*p* < 0.001) and AUC of insulin on the OGTT 0–180 min (*p* = 0.030) and HOMA-IR (*p* < 0.001). The Stumvoll insulin resistance index was also worse (*p* = 0.032). Higher levels of white blood cells (*p* < 0.001), CRP (*p* < 0.001), and complement fractions C3 and C4 (*p* = 0.001 and *p* = 0.004, respectively) emerged.

There was no difference, however, in the levels of ACTH and cortisol, while the presence of reduced levels of IGF-1 was confirmed in patients with obesity (*p* < 0.001).

Evaluating the population of childbearing age, no differences emerged in the levels of gonadotropins, estradiol, progesterone, and androgens.

Considering the ultrasound picture, the thickness of the subcutaneous tissue, as expected, was significantly greater in all points evaluated on the lower limb (*p* < 0.001 in all points). The thickness of the skin was also greater in all points except the upper and lower lateral third of the leg.

As a final aspect, regarding the comorbidities reported in Table 6, we found greater prevalence in patients with obesity of venous disease (79.3 vs. 67.3%, *p* = 0.009), depressed mood (57.6 vs. 37.3%, *p* < 0.001), urinary incontinence (43.2% vs. 16.2%, *p* <0.001), a history of fractures (39.3 vs. 22.8%, P0 = 0.002) and gastric disorders (41.4 vs. 27.6%, *p* = 0.006), hypothyroidism (29.3 vs. 18.2%, *p* = 0.010) and hypertension (26.1 vs. 6.9%, *p* < 0.001), gestational diabetes (10.5 vs. 3.1%, *p* = 0.049), metabolic syndrome (16.3 vs. 1.9%, *p* < 0.001), alterations of glucose metabolism (50 vs. 24.8%, *p* < 0.001), insulin resistance (46.7 vs. 21.3%, *p* < 0.001), and impaired fasting glucose (9.8 vs. 0.9%, *p* = 0.009). No other differences emerged in the prevalence of the comorbidities listed in the table.

## 3. Materials and Methods

This was a retrospective observational study on 360 female patients diagnosed with stage 1, 2, or 3 lower limb lipedema, consecutively evaluated at our institute starting in June 2021. The diagnosis was based on the clinical findings of the characteristic signs and symptoms of the disease.

The lymphological examination consisted of a systemic evaluation of the clinical aspects functional for diagnosis, the definition of the clinical stage, and phenotyping: the entity and distribution of the adipose tissue; the appearance and conformation of the lower and upper limbs, abdomen, and back; the appearance and consistency of the skin and subcutis; the research and characterization of subcutaneous nodulations and evaluation of tissue laxity, signs of edema (fovea, increased consistency of skin folds especially on the backs of feet and hands, presence of Stemmer’s sign) and signs and symptoms suggestive of alterations of the microcirculation and venous diseases (skin temperature, skin discoloration and trophic changes of the skin, telangiectasias, presence of telangiectasias, superficialization and appearance of venous vessels, presence of varicose and venous ectasias), and the presence of capillary fragility (ecchymoses, hematomas). Other aspects evaluated were the presence of ankle and wrist cuffs, a dimple in the medial supramalleolar region, and suprapatellar and malleolar fat pads (Figure 3).

A fundamental aspect evaluated in all patients was the presence of pain evoked by pinching the adipose tissue at several points of the lower limbs and on the abdomen, back, and upper limbs. The presence of pain, together with the other signs and symptoms and the ultrasound evaluation, should be considered an integral part of the clinical evaluation and in support of the diagnosis of lipedema. The presence of pain when pinching the skin in different points of the lower limb, upper limb, abdominal, and back was evaluated with a Verbal Rating Scale (VRS). The VRS used in our study was a 5-point Likert-type scale with the following verbal descriptors: no pain, mild, moderate, severe, and very severe pain [62]. The VRS was chosen because it is preferred by patients for its simplicity compared to the Visual Analogue Scale (VAS) and Numerical Rating Scale (NRS). It is considered more realistic for the definition of pain perceived. One of the first reviews comparing the three pain scales (VAS, VRS, and NRS) was published by Williamson and Hoggart in 2005, and they reported that all three scales were valid, reliable, and appropriate for use [63,64]. For the evaluation of pain intensity, we used a numerical value from 0 to 4 (0 = no pain, 1 = mild pain, 2 = moderate pain, 3 = severe pain, 4 = very severe pain) and recorded these values for each point assessed.

For the lower limb, we evaluated the presence of pain in 7 points: the posterior middle third of the leg (immediately below the end of the gastrocnemius muscle), the medial upper third of the leg (medial to the prominence of the tibial tuberosity, called Simarro’s sign), the medial and lateral lower third of the thigh (3–5 above the knee), the lateral upper third of the thigh (immediately below the trochanter), the medial upper third of the thigh (approximately 5 cm below the inguinal crease), and the lower abdomen (laterally and directly below the navel) (Figure 4). The compression of the tissue fold was carried out with the whole hand, compressing the liftable subcutaneous tissue with maximum pressure between the thenar eminence and the four fingers of the hand.

At the level of the upper limb, the pain was assessed with the arm raised at 90 degrees from the trunk, on the lower edge, carrying out the test at several points from the distal third to the proximal third of the arm and forearm (recording, respectively, the highest score obtained for the arm and the forearm) and posteriorly, on the lower edge of the subscapularis muscle, arm and forearm (Figure 4). The tissue compression at these points was carried out by lifting the fold between the four fingers and the thumb.

We obtained three scores: a lower limb score calculated with the mathematical mean of the single values found in the 7 points described for the lower limbs, an upper limb score calculated with the mathematical mean of the values obtained from the three points described for the upper limbs, and a total score, calculated with the mathematical mean of the 10 points evaluated.

Finally, joint hypermobility was evaluated with the Beighton score [65].

The differential diagnosis, particularly with lymphedema and severe venous disease associated with phlebedema, was supported not only by the clinical evaluation but also with a venous color Doppler ultrasound and with the ultrasound study of the superficial tissues of the lower limbs (with assessment of the dermo-epidermis complex, assessment of the presence of lymphatic lacunae and other signs of lymphedema).

All patients underwent medical history, anthropometric (weight, height, waist, and hip circumference), and general clinical evaluations with particular attention to clinical signs of endocrinopathy (hirsutism, acanthosis nigricans, acne, alopecia, presence of dorsal hump, and striae rubra). For each patient, essential medical history data were collected to evaluate the presence or absence of other comorbidities.

The laboratory tests performed (but not available for all patients evaluated) included general tests (total blood count, liver, and kidney function), C-reactive protein (CRP), C3 and C4 complement fractions, fasting blood sugar and insulin, glycated hemoglobin and oral glucose tolerance test (OGTT), thyroid function (thyroid stimulating hormone (TSH), free triiodothyronine (FT3), free thyroxine (FT4), and autoantibodies anti-thyroglobilin and anti-thyroperoxidase and gonadotropins), 25-OH vitamin D, and insulin-like growth factor-1 (IGF-1). We also performed hormonal assays of the pituitary–ovarian and pituitary–adrenal axis: lutenizing hormone (LH), follicle stimulating hormone (FSH), estradiol, progesterone, adrenocorticotropic hormone (ACTH), testosterone, androstenedione, dehydroepiandrosterone sulfate (DHEAS), dehydroepiandrosterone (DHEA), sex hormone binding globulin (SHBG), 17α-hydroxyprogesterone: these hormones were evaluated in the follicular phase in all fertile patients and at least two months after the discontinuation of any estrogen-progestins, or randomly, in the case of amenorrhea or menopause.

Most patients also underwent an ultrasound study of the areas affected by lipedema to measure the skin and subcutaneous tissue’s thickness. At the level of the lower limbs, measurements were carried out at 9 different points, 5 of which corresponded to the pain detection points already described above, except for the rear calf area and lower abdomen: the medial upper third of the leg, medial and lateral lower third of the thigh, lateral upper third of the thigh, and medial upper third of the thigh. In addition to the locations in common with pain detection, the measurement was also carried out at the level of the lower medial third of the leg (3–5 cm above the medial malleolus), at the level of the lower lateral third of the leg (3–5 cm above the lateral malleolus), and at the upper lateral third of the leg and upper anterior third of the thigh (approximately 5 cm below the inguinal crease in the anterior region of the thigh) (Figure 4). The measurement was performed using a high-frequency linear probe of 8–14 MHz (SonoScape X3, SonoScape Medical Corp, Shenzhen, China), keeping the probe perpendicular to the skin and without applying pressure to the underlying tissue. For each point, two measurements were carried out: the thickness of the skin measured from the skin surface to the lower edge of the epidermis and dermis complex, and the thickness of the suprafascial superficial adipose tissue, measured from the surface of the skin to the muscle fascia (Figure 5).

## 4. Conclusions

This work aimed to describe the clinical, instrumental, and blood test results of a large population of women affected by lipedema of the lower limbs, given that data in the literature are still limited. There are no previous study reports on this in the Italian population. We have already reported preliminary data on clinical signs, symptoms, comorbidities, and endocrine-metabolic findings on a smaller population (about 200 patients) [66,67]. Now, we have data on over 350 cases, and although most of the results were confirmed, differences emerged in some cases, and this was probably also due to the need for more homogeneity of the population in many aspects, not merely clinical ones. For the diagnosis and definition of the severity of the disease, it is essential to evaluate the history, clinical and ultrasound aspects, and specific blood tests together based on the knowledge obtained through recent studies. This multifactorial assessment of patients is also essential for the phenotyping of this population, which is undoubtedly composed of clinical pictures with different genetic and pathogenic bases, considering the high prevalence in the general population and the diversity of cases observed in daily routine. For example, evaluating the differences in the subpopulations of patients affected by lipedema with different onset times related to the hormonal profile due to a different history or clinical picture or based on ultrasound findings may also be helpful. It could also help to understand the pathogenesis better. However, phenotyping could also help to guide the choice of other tests or examinations for the patient and the choice of conservative and even surgical treatments. From our first evaluation of the data, it is clear that a more severe clinical stage, a higher BMI, or the concomitant involvement of the upper limbs suggests the higher prevalence of comorbidities and a worse endocrine–metabolic profile. These aspects, even if taken separately, already help to identify patients who need to be studied not only from a lymphological and vascular point of view but also from an endocrine–metabolic point of view. Furthermore, they support the need to structure a set of valuable indications for a systematic and objective clinical evaluation, including the evaluation of pain (not only subjective) and possibly an ultrasound evaluation. This approach is fundamental to studying the patient “in toto” to obtain a definition of the clinical picture that is as complete as possible, in order to formalize a therapeutic assistance path inherent to the “state” of global health of the patient, rather than focusing only on the state of health as a loco-regional aspect of the pathology.

## Figures and Tables

**Figure 1 ijms-25-01599-f001:**
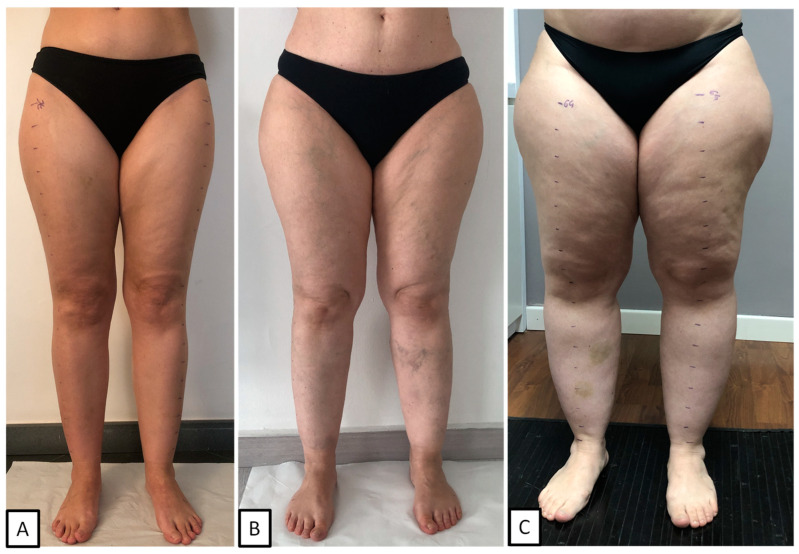
Staging of lipedema. The figure shows the three clinical stages of lipedema: stage 1 (**A**), stage 2 (**B**), and stage 3 (**C**).

**Figure 2 ijms-25-01599-f002:**
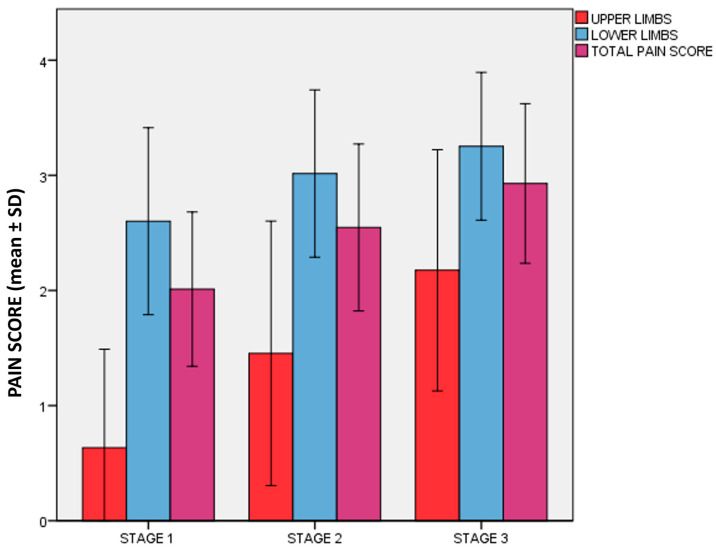
Pain score in the clinical stages. Figure 2 shows the mean of the lower limb pain score, upper limb pain score, and total pain score in the three stages. Pain was quantified using a Verbal Rating Scale with values from 0 to 4 (0 = no pain, 4 = maximum pain), applied to each specific site (as described in the text), evaluating three different scores: a lower limb pain score, an upper limb pain score, and a total pain score. The mean of all three pain scores increased with the severity of the clinical stage (*p* < 0.001 for all). For the lower pain score, the difference was significant when comparing stage 1 with stages 2 (*p* < 0.001) and 3 (*p* < 0.001); it was not significant when comparing stage 2 with stage 3 (*p* = 0.0125). The difference was always significant in the comparison of each stage for the upper limb and total pain scores (*p* < 0.001 for each comparison).

**Figure 3 ijms-25-01599-f003:**
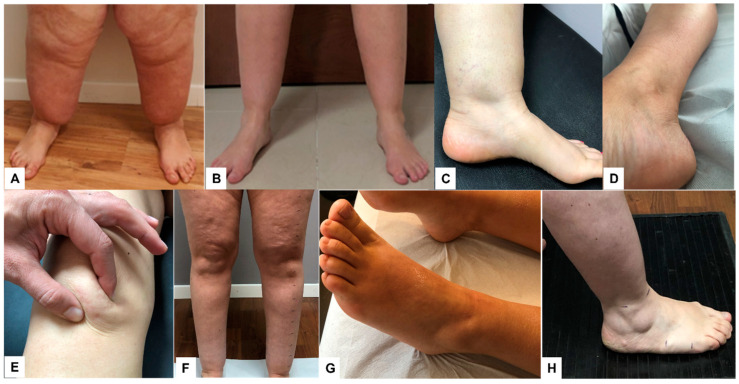
Clinical signs of lipedema. The figure shows some of the characteristic clinical signs of lipedema: ankle cuff (**A**,**B**), malleolar dimple (**C**,**D**), suprapatellar fat pad (**E**,**F**), malleolar fat pad (**G**,**H**).

**Figure 4 ijms-25-01599-f004:**
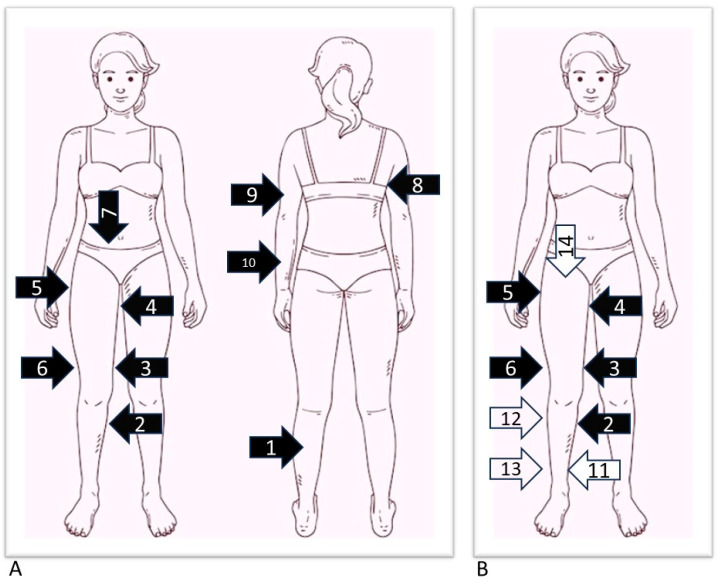
(**A**,**B**) Points of pain detection (**A**) and adipose tissue thickness evaluation via ultrasound (**B**). (**A**) The figure shows the points of pain detection of lower and upper limbs: posterior middle third of the leg (1), medial upper third of the leg (2), medial and lateral lower third of the thigh (3 and 4), lateral upper third of the thigh (5), medial upper third of the thigh (6) and lower abdomen (7), lower edge of the subscapularis muscle (8), arm and forearm (9 and 10). (B) The figure shows the adipose tissue points where the thickness was assessed by ultrasound; some of these are in common with the pain detection points: medial upper third of the leg (2), medial and lateral lower third of the thigh (3 and 4), lateral upper third of the thigh (5), medial upper third of the thigh (6). Furthermore, the thickness of the adipose tissue was measured at four other points: lower medial third of the leg (11), lower lateral third of the leg (13), upper lateral third of the leg (12), upper anterior third of the thigh (14).

**Figure 5 ijms-25-01599-f005:**
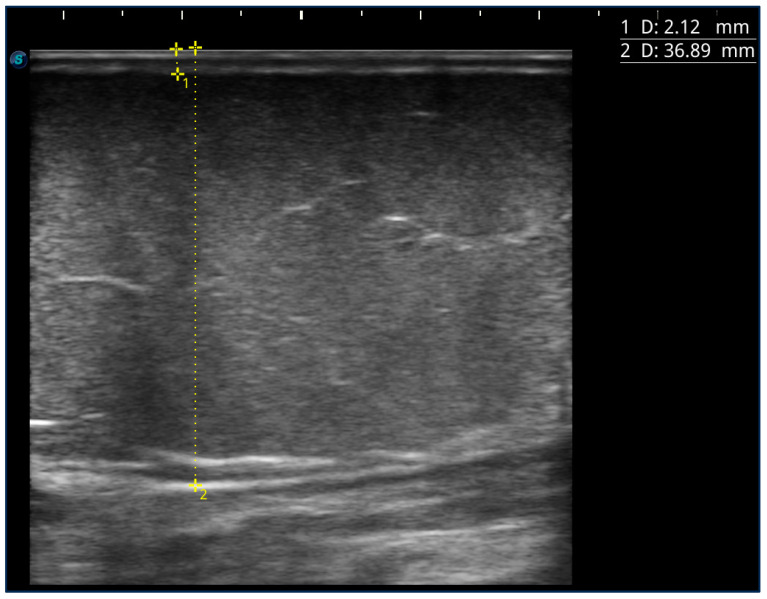
Ultrasound measurement of skin (dashed line 1) and adipose tissue thickness (dashed line 2): the thickness of the skin was measured from the skin surface to the lower edge of the epidermis and dermis complex, and the thickness of the suprafascial superficial adipose tissue was measured from the skin surface to the subcutaneous transition structure (fascia). This image is related to the lower medial third of the right leg, performed using a high-frequency linear probe (8–14 MHz), keeping the probe perpendicular to the skin and without applying pressure to the underlying tissue.

**Table 1 ijms-25-01599-t001:** Anthropometric and clinical data.

VARIABLE	ALL	STAGE 1	STAGE 2	STAGE 3	*p* STAGE	WITHOUT UP	WITH UL	*p* UP
Age (yrs)	40.4 ± 11.8 (360)	37.7 ± 12.4 (143)	41.6 ± 10.9 (144) *	43.4 ± 11.6 ^§^ (73)	0.001	38.42 ± 12.6 (163)	42.1 ± 10.9 (197)	0.003
Body weight (kg)	78.4 ± 17.1 (360)	66.2 ± 9.38 (143)	79.4 ± 12.9 (144) *	100 ± 19.2 ^§^ (73)	<0.001	70.9 ± 13.5 (163)	84.6 ± 19.2 (197)	<0.001
Body mass index (kg/m^2^)	29.4 ± 6.83 (360)	24.7 ± 3.21 (143)	29.7 ± 4.52 (144) *	31.2 ± 7.07 ^§+^ (73)	<0.001	26.4 ± 4.86 (163)	31.9 ± 7.23 (197)	<0.001
Waist circumference (cm)	94.5 ± 14.0 (338)	86.1 ± 9.13 (135)	96.3 ± 12.0 (135) *	107 ± 14.6 ^§+^ (68)	<0.001	88.0 ± 11.3 (157)	100 ± 13.7 (181)	<0.001
Hip circunference (cm)	116 ± 13.0 (338)	107 ± 7.28 (135)	117 ± 9.75 (135) *	131 ± 12.3 ^§+^ (68)	<0.001	111 ± 11.0 (157)	120 ± 13.2 (181)	<0.001
WHR	0.820 ± 0.078 (338)	0.810 ± 0.068 (135)	0.820 ± 0.076 (135)	0.820 ± 0.096 (135)	NS	0.790 ± 0.064 (157)	0.840 ± 0.083 (181)	<0.001
WHtR	0.580 ± 0.088 (338)	0.530 ± 0.055 (135)	5.590 ± 0.075 135) *	0.670 ± 0.091 ^§+^ (68)	<0.001	0.540 ± 0.070 (157)	0.620 ± 0.087 (181)	<0.001
Hirsutism (%)	5.30 (19/359)	5.6 (8/143)	5.6 (8/144)	4.2 (3/72)	NS	6.7 (11/163)	4.1 (8/196)	NS
Acanthosis (%)	32.3 (116/359)	18.2 (26/143)	33.3 (48/144) *	58.3 (42/72) ^§+^	<0.001	22.1 (36/163)	40.8 (80/196)	<0.001
Dorsal hump (%)	17.4 (55/316)	5.4 (7/129)	21.0 (25/119) *	33.8 (23/68) ^§+^	<0.001	2.1 (3/140)	29.5 (52/176)	<0.001
Acne (%)	4.40 (16/360)	7.7 (11/143)	1.4 (2/144) *	4.1 (3/73)	0.034	6.1 (10/163)	3.0 (6/197)	NS
Alopecia (%)	1.90 (7/360)	2.1 (3/143)	1.4 (2/144)	2.7 (2/73)	NS	3.7 (6/163)	0.5 (1/197)	0.090
Age of onset (yrs)	13.4 ± 9.49 (360)	13.9 ± 9.85 (143)	13.5 ± 9.40 * (144)	12.4 ± 8.98 (73)	NS	12.7 ± 7.90 (163)	14.0 ± 10.6 (197)	NS
Duration of the disease (yrs)	27.0 ± 13.1 (360)	23.9 ± 13.5 (143)	28.1 ± 12.5 * (144)	31 ± 12.5 ^§^ (73)	<0.001	25.7 ± 13.1 (163)	28.1 ± 13.1 (197)	NS
Maximum weight reached (kg)	85.8 ± 22.3 (360)	71.0 ± 10.7 (143)	89.5 ± 20.2 * (144)	108 ± 22.3 ^§+^ (73)	<0.001	78.1 ± 17.7 (163)	98.2 ± 23.7 (197)	<0.001

The table shows the anthropometric and clinical data in the entire population affected by lipedema (ALL) and in the population divided into the three clinical stages (STAGE 1, STAGE 2, STAGE 3) and in the two groups of patients without involvement of the upper limbs (WITHOUT UP) and with upper limb involvement (WITH UL). The data are expressed with mean ± standard deviation or in % (n° cases/total). *p* values compare the three clinical stages (*p* STAGE) and the patients with and without involvement of the upper limbs (*p* UL). ANOVA evaluated the comparison between groups for the continuous variables and Pearson’s chi-square was used to compare the prevalence of the discrete variables in the different groups. NS: no significant statistical difference; * = *p* value < 0.05 stage 1 vs. 2; ^§^ = *p* value < 0.05 stage 1 vs. stage 3; ^+^ = *p* value < 0.05 stage 2 vs. stage 3.

**Table 2 ijms-25-01599-t002:** Symptoms and clinical lymphological signs.

VARIABLE	ALL	STAGE 1	STAGE 2	STAGE 3	*p STAGE*	WITHOUT UP	WITH UL	*p* UL
Spontaneous pain (%)	70.0 (252/360)	67.1 (96/143)	72.9 (105/144)	69.9 (51/73)	NS	67.5 (110/163)	72.1 (142/197)	NS
Any symptoms (pain, swelling, heaviness) (%)	92.5 (333//360)	91.6 (131/143)	94.4 (136/144)	90.4 (66/73)	NS	92.6 (151/163)	92.4 (182/197)	NS
Ease of bruising (%)	80.8 (291/360)	79.7 (114/143)	79.9 (115/144)	84.9 (62/73)	NS	77.3 (126/163)	83.3 (165/197)	NS
Subcutaneous nodules (%)	98.9 (351/348)	97.2 (138/142)	100 (142/142)	100 (71/71)	0.048	97.5 (158/162)	100 (193/193)	0.042
Suprapatellar fat pads (%)	93.8 (331/353)	87.9 (124/141)	91.2 (137/141) *	98.6 (70/71) ^§^	0.001	91.3 (147/161)	95.8 (184/192)	NS
Medial retromalleolar dimple (%)	86.7 (294/339)	81.8 (112/137)	88.1 (119/135)	94.0 (63/67) ^§^	0.043	84.9 (129/152)	88.2 (165/187)	NS
Perimalleolar fat pads (%)	87.2 (307/352)	80.0 (112/140)	92.2 (130/141) *	91.5 (65/71) ^§^	0.004	82.5 (132/160)	91.1 (175/192)	0.012
Telangiectasia (%)	71.2 (245/344)	59.0 (82/139)	76.5 (104/136) *	85.5 (59/69) ^§^	<0.001	66.0 (105/159)	75.7 (140/185)	0.032
Ankle cuff sign (%)	62.7 (224/357)	51.7 (74/143)	64.8 (92/142) *	80.6 (58/72) ^§+^	<0.001	53.1 (86/162)	70.8 (138/195)	0.001
Anterior pretibial fovea (%)	14.7 (52/301)	5.6 (8/142)	17.1 (24/140) *	28.2 (20/71) ^§+^	<0.001	6.2 (10/162)	22.0 (42/191)	<0.001
Increased back foot fold (%)	9.3 (33/356)	4.2 (6/143)	13.4 (19/142) *	11.3 (8/71) ^§^	0.023	6.24 (10/162)	11.9 (23/194)	0.047
Foot edema (back and toes) (%)	2.8 (10/356)	1.4 (2/143)	3.5 (5/142)	4.2 (3/71)	NS	1.2 (2/162)	4.1 (8/194)	NS
Positive Stemmer’s sign (%)	1.7 (6/354)	0 (0/140)	2.8 (4/144)	2.7 (2/73)	NS	0.6 (1/163)	2.5 (5/197)	N S
Joint hypermobility (%)	31.5 (73/232)	33.3 (28/84)	27.0 (27/100)	37.5 (18/48)	NS	23.3 (30/129)	41.7 (43/103)	0.002
Pain on pinching								
Any point (lower limb, upper limb, and trunk) (%)	99.4 (358/360)	99.3 (142/143)	99.3 (143/144)	100 (73/73)	NS	99.4 (162/163)	99.5 (196/197)	NS
Posterior middle third of the leg (%)	96.0 (333/347)	93.6 (131/140)	97.1 (134/138)	98.6 (68/69)	NS	97.5 (154/158)	94.7 (179/189)	NS
Medial upper third of the leg (Simarro) (%)	97.2 (348/358)	95.8 (137/143)	97.9 (140/143)	98.6 (71/72)	NS	98.1 (159/162)	96.4 (189/196)	NS
Medial lower third of the thigh (%)	98.0 (339/346)	96.4 (135/140)	98.6 (136/138)	100 (68/68)	NS	98.7 (154/156)	97.4 (185/190)	NS
Medial upper third of the thigh (%)	95.7 (331/346)	92.1 (129/140)	97.8 (135/138) *	98.5 (97/68)	0.029	93.6 (146/156)	97.4 (185/190)	NS
Lateral upper third of the thigh (%)	90.5 (314/347)	87.1 (122/140)	92.0 (127/138)	94.2 (65/69)	NS	84.1 (132/157)	95.8 (182/190)	<0.001
Lateral lower third of the thigh (%)	96.2 (333/346)	94.3 (132/140)	97.8 (135/138)	97.1 (66/68)	NS	94.9 (148/156)	97.4 (185/190)	NS
Lower abdomen (%)	28.6 (99/346)	21.4 (30/140)	29.7 (41/138)	41.2 (28/68) ^§^	0.012	21.8 (34/156)	34.2 (65/190)	0.007
Lower edge of the subscapularis muscle (%)	57.1 (197/345)	35.7 (50/140)	65.0 (89/137) *	85.3 (58/68) ^§+^	<0.001	21.3 (33/155)	86.3 (164/190)	<0.001
Arm (%)	61.2 (213/348)	40.0 (56/140)	67.4 (93/138) *	91.4 (64/70) ^§+^	<0.001	19.4 (30/155)	94.8 (183/193)	<0.001
Forearm (%)	22.1 (77/348)	7.9 (11/140)	23.9 (33/138) *	47.1 (33/70) ^§+^	<0.001	2.6 (4/155)	37.8 (73/193)	<0.001

Table 2 shows the prevalence of symptoms and clinical lymphological signs in the entire population affected by lipedema (ALL), in the population divided into the three clinical stages (STAGE 1, STAGE 2, STAGE 3), and in the two groups of patients without involvement of the upper limbs (WITHOUT UL) and with upper limb involvement (WITH UL). The data are expressed as % of cases (n° cases/total). A comparison between the three clinical stages (*p* STAGE) and the groups with/without upper limb involvement (*p* UL) is reported. Pearson’s chi-square was used to evaluate the comparison between groups. NS: no significant statistical difference; * = *p* value < 0.05 stage 1 vs. 2; ^§^ = *p* value < 0.05 stage 1 vs. stage 3. ^+^ = *p* value < 0.05 stage 2 vs. stage 3.

**Table 3 ijms-25-01599-t003:** Skin and tissue thicknesses.

VARIABLE	ALL	STAGE 1	STAGE 2	STAGE 3	*p* STAGE	*p* 1 vs. 2	*p* 1 vs. 3	*p* 2 vs. 3
Thickness of Skin								
lower medial third of the leg (mm)	2.01 ± 0.323	1.95 ± 0.278	1.95 ± 0.435	2.19 ± 0.407	0.006	NS	0.011	0.012
the medial upper third of the leg (mm)	1.85 ± 0.265	1.79 ± 0.245	1.83 ± 0.236	1.99 ± 0.306	0.015	NS	0.014	NS
medial lower third of the thigh (mm)	1.878 ± 0.351	1.76 ± 0.267	1.87 ± 0.361	2.07 ± 0.379	0.003	NS	0.002	NS
medial upper third of the thigh (mm)	1.92 ± 0.374	1.74 ± 0.284	1.93 ± 0.294	2.21 ± 0.452	<0.001	NS	<0.001	0.005
upper anterior third of the thigh (mm)	1.68 ± 0.293	1.61 ± 0.261	1.63 ± 0.239	1.91 ± 0.339	<0.001	NS	<0.001	0.001
lateral upper third of the thigh (mm)	2.72 ± 0.417	2.66 ± 0.389	2.74 ± 0.389	2.801 ± 0.524	NS	NS	NS	NS
lateral lower third of the thigh (mm)	2.075 ± 0.306	1.99 ± 0.280	2.09 ± 0.330	2.20 ± 0.266	0.033	NS	0.029	NS
upper lateral third of the leg (mm)	1.92 ± 0.276	1.92 ± 0.230	1.89 ± 0.290	1.98 ± 0.325	NS	NS	ns	NS
lower lateral third of the leg (mm)	1.83 ± 0.413	1.84 ± 0.207	1.72 ± 0.431	2.02 ± 0.567	<0.001	NS	ns	0.023
Thickness of Adipose Tissue								
lower medial third of the leg (mm)	22.1 ± 7.62	17.6 ± 4.45	21.8 ± 5.35	29.7 ± 9.02	<0.001	0.010	<0.001	<0.001
medial upper third of the leg (mm)	22.7 ± 8.71	18.3 ± 6.61	23.8 ± 8.30	28.3 ± 9.00	<0.001	0.009	<0.001	NS
medial lower third of the thigh (mm)	35.3 ± 10.5	28.9 ± 6.17	35.4 ± 8.68	45.7 ± 10.9	<0.001	0.003	<0.001	<0.001
medial upper third of the thigh (mm)	33.2 ± 12.7	24.3 ± 6.76	33.4 ± 7.96	47.4 ± 13.6	<0.001	<0.001	<0.001	<0.001
upper anterior third of the thigh (mm)	17.0 ± 8.44	12.3 ± 4.021	16.7 ± 6.47	26.1 ± 10.4	<0.001	0.019	<0.001	<0.001
lateral upper third of the thigh (mm)	79.7 ± 24.0	64.1 ± 14.9	82.1 ± 20.5	104 ± 24.0	<0.001	0.001	<0.001	0.001
lateral lower third of the thigh (mm)	25.0 ± 10.3	18.2 ± 6.6	24.5 ± 7.1	37.8 ± 8.6	<0.001	0.001	<0.001	<0.001
upper lateral third of the leg (mm)	14.3 ± 6.5	12.1 ± 7.60	14.3 ± 4.67	18.4 ± 5.75	0.001	NS	0.001	0.043
lower lateral third of the leg (mm)	12.1 ± 5.7	9.70 ± 4.21	11.3 ± 4.25	17.1 ± 6.84	<0.001	NS	<0.001	<0.001

The table shows the thickness of skin and adipose tissue evaluated in a subpopulation of 102 patients diagnosed with type 3 lipedema who had not previously undergone liposuction, had not followed diets or had body weight reductions > 10% of weight, and had blood tests available. The measurements refer to the entire subpopulation (ALL) and the population divided into the three clinical stages (STAGE 1, STAGE 2, STAGE 3). The data are expressed as the mean ± standard deviation. The table also shows the *p* value between the stages (*p* STAGE) and the *p* value of the comparison between the individual stages: comparison between stages 1 and 2 (*p* 1 vs. 2), comparison between stages 1 and 3 (*p* 1 vs. 3), and comparison between stage 2 and 3 (*p* 2 vs. 3). ANOVA was used to evaluate the comparison between groups.

**Table 4 ijms-25-01599-t004:** Inflammatory markers and hormonal and metabolic characteristics.

VARIABLE	ALL	STAGE 1	STAGE 2	STAGE 3	*p STAGE*	WITHOUT UP	WITH UL	*p* UP
White blood cells (10^9^/L)	5.90 ± 1.76 (207)	5.72 ± 1.67 (80)	5.92 ± 1.84 (83)	6.17 ± 1.77 (44)	NS	5.64 ± 1.69 (106)	6.17 ± 1.80 (101)	0.029
CRP (mg/L)	2.31 ± 2.70 (159)	1.38 ± 1.29 (67)	1.94 ± 1.69 (60)	4.93 ± 4.35 (32) ^§#^	<0.001	1.422 ± 1.50 (78)	3.16 ± 3.27 (81)	<0.001
Total cholesterol (mg/dL)	191 ± 34.0 (206)	109 ± 35.8 (81)	192 ± 34.2 (81)	193 ± 30.8 (44)	NS	193 ± 37.2 (106)	190 ± 30.6 (100)	NS
HDL cholesterol (mg/dL)	63.9 ± 13.7 (201)	65.6 ± 13.6 (81)	65.3 ± 13.1 (76)	58.2 ± 13.6 (44) ^§#^	0.007	66.7 ± 14.0 (108)	60.6 ± 12.5 (93)	0.001
Triglycerides (mg/dL)	78.9 ± 45.2 (200)	74.4 ± 51.4 (79)	75.5 ± 64.6 (77)	91.0 ± 48.6 (44)	NS	70.3 ± 30.0 (102)	87.9 ± 55.7 (98)	0.006
LDL cholesterol (mg/dL)	90.1 ± 10.2 (188)	88.6 ± 8.6 (74)	89.6 ± 11.3 (73)	93.5 ± 10.2 (41)	NS	111 ± 30.5 (98)	112 ± 27.3 (90)	NS
TSH (uIU /mL)	2.25 ± 1.37 (217)	2.21 ± 1.18 (84)	2.27 ± 1.35 (89)	2.30 ± 1.73 (44)	NS	2.30 ± 1.38 (113)	2.20 ± 1.36 (104)	NS
Prolactin (ng/mL)	15.4 ± 8.83 (160)	16.2 ± 8.62 (62)	15.7 ± 9.49 (63)	13.4 ± 7.82 (35)	NS	16.1 ± 8.10 (84)	14.6 ± 9.56 (76)	NS
ACTH (pg /mL)	25.1 ± 18.3 (164)	27.1 ± 22.6 (66)	22.4 ± 11.1 (63)	25.9 ± 19.7 (35)	NS	27.1 ± 21.6 (86)	22.8 ± 13.6 (78)	NS
Cortisol (ng/mL)	148 ± 57.3 (176)	151 ± 48.3 (68)	139 ± 38.9 (68)	157 ± 88.6 (40)	NS	179 ± 48.1 (89)	146 ± 65.6 (87)	NS
IGF-1 (ug /L)	162 ± 74.0 (160)	190 ± 87.6 (66)	150 ± 56.0 (56) *	129 ± 51.3 (38) ^§^	<0.001	181 ± 84.6 (85)	141 ± 52.9 (75)	0.001
Vitamin D25-OH (ng/mL)	25.3 ± 11.2 (195)	26.2 ± 10.5 (78)	25.9 ± 12.7 (74)	22.7 ± 9.2 (43)	NS	26.2 ± 11.1 (100)	24.3 ± 11.2 (95)	NS
C3 complement fraction (mg/dL)	113 ± 24.5 (46)	107 ± 20.1 (19)	111 ± 25.6 (17)	131 ± 23.9 (10) ^§^	0.027	93.4 ± 17.8 (10)	23.3 ± 3.89 (36)	0.002
C4 complement fraction (mg/dL)	28.4 ± 10.3 (46)	25.9± 9.52 (19)	26.7 ± 10.2 (17)	36.0 ± 9.2 (10) ^§^	0.027	18.4 ± 3.92 (10)	31.2 ± 9.85 (36)	<0.001
HBA1c (mmol/mol)	34.2 ± 4.3 (189)	34.0 ± 3.8 (76)	33.5 ± 4.39 (73)	35.6 ± 4.68 (40) ^#^	0.043	32.3 ± 3.96 (99)	34.0 ± 4.59 (90)	NS
HOMA-IR	2.12 ± 1.51 (202)	1.75 ± 1.24 (76)	2.01 ± 1.54 (80)	2.92 ± 1.60 (46) ^§#^	<0.001	1.88 ± 1.43 (101)	2.36 ± 1.56 (101)	0.023
Basal glucose OGTT (mg/dL)	90.1 ± 10.2 (231)	88.6 ± 8.6 (89)	89.6 ± 11.3 (91)	93.5 ± 10.2 (51) ^§^	0.020	89.4 ± 9.77 (116)	90.7 ± 10.7 (115)	NS
Glucose 60 min OGTT (mg/dL)	118 ± 38.5 (146)	109 ± 33.3 (57)	121 ± 44.3 (53)	127 ± 35.3 (36)	NS	116 ± 38.4 (71)	120 ± 38.8 (75)	NS
Glucose 120 min OGTT (mg/dL)	96.0 ± 27.1 (170)	89.0 ± 22.7 (67)	96.3 ± 22.8 (63)	107 ± 35.7 (40) ^#^	0.003	93.0 ± 23.2 (83)	98.8 ± 30.3 (87)	NS
Basal insulin OGTT (uIU/mL)	9.33 ± 6.07 (207)	7.84 ± 5.32 (78)	8.94 ± 6.14 (83)	12.5 ± 6.08 (46) ^§#^	<0.001	8.39 ± 5.90 (103)	10.2 ± 6.12 (104)	0.027
Insulin 60 min OGTT (uIU/mL)	64.5 ± 50.0 (137)	53.3 ± 39.2 (51)	69.6 ± 61.0 (53)	73.7 ± 43.0 (33)	NS	54.74 ± 38.8 (62)	72.6 ± 55.63 (75)	0.037
Insulin 120 min OGTT (uIU/mL)	52.72 ± 60.0 (143)	45.7 ± 37.4 (54)	54.8 ± 82.5 (55)	60.5 ± 44.0 (34)	NS	44.9 ± 33.5 (65)	59.2 ± 74.9 (78)	NS
Glucose AUC 0–120 min OGTT	13,620 ± 2946 (119)	12,930 ± 2500 (49)	13,854 ± 3254 (39)	14,417 ± 3035 (31)	NS	13,309 ± 2646 (54)	13,880 ± 3171 (65)	NS
Glucose AUC 0–180 min OGTT	18,744 ± 3671 (102)	18,241 ± 3626 (40)	19,052 ± 4064 (35)	19,090 ± 3220 (27)	NS	18,634 ± 3696 (49)	18,845 ± 3680 (53)	NS
Insulin AUC 0–120 min OGTT	6470 ± 5463 (115)	5195 ± 3432 (46)	7205 ± 7565 (40)	7478 ± 4290 (29)	NS	5412 ± 3222 (50)	7284 ± 6609 (65)	NS
Insulin AUC 0–180 min OGTT	8367 ± 5224 (86)	7160 ± 4376 (30)	8358 ± 5406 (31)	9825 ± 5740 (25)	NS	7560 ± 4335 (40)	9069 ± 5846 (46)	NS
Stumvoll 0–120 index OGTT	0.120 ± 0.29 (141)	0.120 ± 0.019 (54)	0.120 ± 0.038 (54) *	0.110 ± 0.024 (33) ^§#^	0.047	0.120 ± 0.021 (65)	0.035 ± 0.004 (76)	NS

The table shows blood tests (ALL) in the population divided into the three clinical stages (STAGE 1, STAGE 2, STAGE 3) and in the two groups of patients with and without upper limb involvement (WITH UL and WITHOUT UP, respectively). Data are means ± standard deviation (n° of cases). The table also shows the OGTT data with fasting (basal) blood glucose and insulin values after 60 min (60 min OGTT) and after 120 min (120 min OGTT) from the oral load of 75 mg of glucose. Also shown are the area under the curve (AUC) values of glycemia and insulin at the loading curve, calculated using the trapezoidal method. AUC was calculated in the OGTT from 0 to 120 min (AUC 0–120) and in the OGTT from 180 min (AUC 0–180). The reported Vitamin D and TSH levels were obtained on the entire population, including patients not under treatment and those under pharmacological treatment. *p* values compare the three clinical stages (*p* STAGE) and the patients with and without involvement of the upper limbs (*p* UL). Analysis by ANOVA. HOMA-IR: calculated homeostasis model assessment insulin resistance; CRP: C-reactive protein, HBA1c: hemoglobin A1c; OGTT: oral glucose tolerance Test; NS: not statistically significant; * = *p* value < 0.05 stage 1 vs. 2; ^§^ = *p* value < 0.05 stage 1 vs. stage 3; ^#^ = *p* value < 0.05 stage 2 vs. stage 3.

**Table 5 ijms-25-01599-t005:** Comorbidities.

Variable	ALL	STAGE 1	STAGE 2	STAGE 3	*p* STAGES	WITHOUT UP	WITH UL	*p* UL
Vitamin D insufficiency	84.6 (193/228)	79.1 (68/86)	88.2 (82/93)	87.8 (43/49)	NS	82.0 (91/111)	87.2 (102/117)	NS
Chronic venous disease	71.9 259/360)	62.2 (89/143)	75.0 (108/144) *	84.9 (62/73) ^§^	0.001	68.7 (112/163)	74.6 (147/197)	NS
Allergies	53.8 (177/329)	43.7 (59/135)	60.8 (79/130) *	60.9 (39/64) ^§^	0.009	45.7 (69/151)	60.7 (108/178)	0.005
Dyslipidemia	47.3 (98/207)	43.2 (35/81)	49.4 (40/81)	51.1 (23/45)	NS	42.1 (45/107)	53.0 (53/100)	NS
Headache	45.4 (157/346)	43.2 (60/139)	47.5 (66/139)	45.6 (31/68)	NS	46.2 (73/158)	44.7 (84/188)	NS
Depression	45.2 (154/341)	39.7 (54/136)	42.8 (59/138)	61.2 (41/67) ^§#^	0.012	36.4 (59/162)	53.1 (95/179)	0.001
Bowel disorders	44.5 (154/346)	49.6 (69/139)	44.2 (61/138)	34.8 (24/69)	NS	43.3 (68/157)	45.5 (86/189)	NS
Obesity	38.9 (140/360)	6.3 (9/143)	44.4 (64/144) *	91.8 (67/73) ^§#^	<0.001	17.2 (28/163)	56.9 (112/197)	<0.001
Chronic autoimmune thyroiditis	35.5 (77/237)	28.0 (26/93)	31.9 (30/94)	42.0 (21/50)	NS	29.2 (33/113)	35.5 (44/124)	NS
Glucose metabolism alterations (all conditions)	34.2 (79/231)	22.5 (20/89)	35.2 (32/91) *	52.9 (27/51) ^§#^	0.001	26.7 (31/116)	41.7 (48/115)	0.012
Gastric disorders	32.9 (113/343)	27.0 (37/137)	37.2 (51/137)	36.2 (25/69)	NS	24.1 (38/158)	40.5 (78/185)	0.001
Menstrual irregularities	32.5 (117/360)	30.8 (44/143)	36.1 (52/144)	28.8 (21/73)	NS	36.8 (60/163)	28.9 (57/197)	NS
Joint hypermobility	31.5 (73/232)	33.3 (28/84)	27.0 (27/100)	37.5 (18/48)	NS	23.3 (30/129)	41.7 (43/103)	0.002
Insulin resistance (HOMA-IR)	30.7 (62/140)	19.7 (15/76)	28.7 (23/80)	52.2 (24/46) ^§#^	0.001	22.8 (23/101)	38.6 (39/101)	0.011
History of spontaneous abortions	30.1 (55/183)	28.1 (16/57)	31.0 (27/87)	30.8 (12/39)	NS	26.3 (20/76)	32.7 (35/107)	NS
History of fractures	29.0 (91/314)	23.3 (30/129)	27.8 (35/126)	44.1 (26/59) ^§#^	0.013	23.4 (34/145)	33.7 (57/169)	0.030
Urinary incontinence	25.8 (59/229)	14.9 (14/94)	28.7 (25/87) *	41.7 (20/48) ^§^	0.002	16.8 (20/119)	35.5 (39/110)	0.001
Ovarian cysts (single or multiple)	23.5 (82/349)	22.9 (32/140)	23.7 (33/139)	24.3 (17/70)	NS	21.7 (35/161)	25.0 (47/188)	NS
Hypothyroidism	22.5 (81/279)	16.1 (23/143)	22.9 (33/144)	34.2 (25/73) ^§^	0.010	23.3 (38/163)	21.8 (43/197)	NS
History of joint sprains/dislocations	20.9 (68/325)	20.3 (27/133)	21.7 (28/129)	20.6 (13/63)	NS	18.8 (29/154)	22.8 (39/171)	NS
Asthma	20.1 (66/329)	18.5 (24/130)	23.1 (31/134)	16.9 (11/65)	NS	18.4 (29/158)	21.6 (37/171)	NS
Polycystic ovary morphology	19.2 (69/291)	18.9 (27/143)	18.8 (27/144)	20.5 (15/73)	NS	18.4 (30/163)	19.8 (39/197)	NS
Polycystic ovary syndrome	17.1 (54/315)	18.5 (23/124)	18.9 (24/127)	10.9 (7/64)	NS	21.1 (30/142)	13.9 (24/173)	NS
Impaired fasting glucose	16.9 (39/231)	9.0 (8/89)	20.9 (19/91) *	23.5 (12/51) ^§^	0.037	13.8 (16/116)	20.0 (23/115)	NS
Uterine fibroids/polyps	15.2 (53/349)	13.5 (19/141)	15.8 (22/139)	17.4 (12/69)	NS	15.4 (25/162)	15.0 (28/159)	NS
Hypertension	14.3 (50/350)	7.1 (10/141)	15.1 (21/139) *	27.1 (19/70) ^§#^	<0.001	5.0 (8/161)	22.2 (42/189)	<0.001
Benign nodules, cysts of the breast	11.6 (35/303)	11.2 (14/125)	8.4 (10/119)	18.6 (11/56)	NS	14.5 (19/131)	9.3 (16/172)	NS
Pregnancy hypertension and eclampsia	11.1 (19/172)	7.4 (4/54)	12.3 (10/81)	13.5 (5/37)	NS	12.7 (9/71)	9.9 (10/101)	NS
Metabolic syndrome	6.6 (16/241)	1.9 (2/103)	6.7 (6/90)	16.7 (8/48) ^§^	0.003	2.3 (3/133)	12.0 (13/108)	0.002
Gestational diabetes	6.4 (11/172)	1.9 (1/54)	11.1 (9/81)	2.7 (1/37)	NS	5.6 (4/71)	6.9 (7/101)	NS
Endometriosis	4.2 (15/345)	4.9 (7/143)	3.5 (5/144)	4.1 (3/73)	NS	5.5 (9/163)	3.0 (6/197)	NS
Impaired glucose tolerance	4.1 (7/170)	1.5 (1/67)	3.2 (2/63)	10.0 (4/40)	NS	1.2 (1/83)	7.0 (6/86)	NS
Diabetes mellitus	0.28 (1/360)	0 (0/143)	0 (0/144)	1.4 (1/73)	NS	0 (0/163)	0.5 (1/197)	NS

The table shows and summarizes the prevalence of comorbidities in all populations (ALL) and the population divided into the three clinical stages (STAGE 1, STAGE 2, STAGE 3) and in the two groups of patients with and without upper limb involvement (WITH UL and WITHOUT UP, respectively). *p* values compare the three clinical stages (*p* STAGE) and the patients with and without involvement of the upper limbs (*p* UL). Data are expressed as % (n° cases/total). Pearson’s chi-square was used to evaluate the comparison between groups. The definition of the individual items is described in the text. NS: not statistically significant; * = *p* value < 0.05 stage 1 vs. 2; ^§^ = *p* value < 0.05 stage 1 vs. stage 3; ^#^ = *p* value < 0.05 stage 2 vs. stage 3.

**Table 6 ijms-25-01599-t006:** Hormonal profile in patients of childbearing age.

VARIABLE	ALL	STAGE 1	STAGE 2	STAGE 3	*p* STAGES	WITHOUT UP	WITH UL	*p* UL
LH (µU/mL)	8.14 ± 8.14 (137)	7.40 ± 3.77 (58)	10.3 ± 11.9 (50)	7.26 ± 5.86 (29)	NS	8.44 ± 8.65 (72)	8.38 ± 7.6 (65)	NS
FSH (vµU /mL)	11.34 ± 14.5 (140)	9.02 ± 6.80 (61)	14.18 ± 21.6 (49)	11.4 ± 10.9 (30)	NS	10.4 ± 13.7 (73)	12.3 ± 15.5 (63)	NS
Progesterone ng/mL	0.512 ± 0.506 (131)	0.575 ± 0.548 (55)	0.493 ± 0.535 (48)	0.422 ± 0.340 (28)	NS	0.570 ± 0.580 (69)	0.450 ± 0.405 (62)	NS
Estradiol (pg/mL)	66.8 ± 60.2 (131)	60.6 ± 43.7 (57)	72.9 ± 69.0 (48)	60.1 ± 74.1 (26)	NS	69.4 ± 71.0 (72)	63.6 ± 44.2 (59)	NS
Androstenedione (ng/dL)	152 ± 96.5 (105)	171 ± 103 (43)	139 ± 103 (41)	138 ± 62.1 (21)	NS	172 ± 102 (55)	129 ± 86.0 (50)	0.022
DHEA	6.32 ± 5.80 (72)	5.96 ± 4.77 (31)	7.02 ± 7.57 (27)	5.78 ± 3.91 (14)	NS	7.04 ± 6.66 (45)	5.12 ± 3.80 (27)	NS
DHEA-S (μg/mL)	1.75 ± 0.925 (126)	1.79 ± 0.94 (56)	1.68 ± 0.86 (44)	1.77 ± 1.01 (26)	NS	1.82 ± 0.977 (68)	1.66 ± 0.861 (58)	NS
17-OH-progesterone (ng/dL)	74.5 ± 37.9 (104)	84.7 ± 37.9 (48)	61.3 ± 36.3 (37) *	74.4 ± 34.4 (19)	0.017	81.1 ± 40.0 (60)	65.5 ± 33.1 (44)	0.038
Total testosterone (ng/mL)	0.29 ± 0.163 (134)	0.304 ± 0.181 (59)	0.273 ± 0.141 (49)	0.290 ± 0.163 (26)	NS	0.284 ± 0.146 (70)	0.297 ± 0.181 (74)	NS
FAI	1.85 ± 1.35 (97)	1.68 ± 1.25 (44)	1.73 ± 1.93 (39)	2.69 ± 179 (14) ^§#^	0.040	1.66 ± 1.086 (59)	2.14 ± 1.65 (38)	0.042
SHBG (mmol/L)	76.5 ± 38.0 (106)	77.7 ± 31.3 (46)	76.2 ± 39.8 (43)	73.8 ± 50.8 (17)	NS	76.1 ± 32.6 (65)	77.1 ± 45.8 (41)	NS

Table 6 shows the hormonal profile (ALL) and is divided into three clinical stages (STAGE 1, STAGE 2, STAGE 3) and two groups of patients with and without upper limb involvement (WITH-UL and WITHOUT-UP, respectively). The data are reported as mean ± standard deviation (n° of cases). *p* values compare the three clinical stages (*p* STAGE) and the patients with and without involvement of the upper limbs (*p* UL). The analysis was performed with ANOVA. LH: luteinizing hormone, FSH: follicle stimulating hormone, DHEA: dehydroepiandrosterone, DHEAS: dehydroepiandrosterone sulfate, FAI: free androgen index, SHBG: sex hormone binding globulin, NS: not statistically significant. * = *p* value < 0.05 stage 1 vs. 2; ^§^ = *p* value < 0.05 stage 1 vs. stage 3; ^#^ = *p* value < 0.05 stage 2 vs. stage 3.

## Data Availability

The derived data supporting the findings of this study are available from the corresponding authors upon reasonable request.

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
