# Peer review of "Observational Study on a Large Italian Population with Lipedema: Biochemical and Hormonal Profile, Anatomical and Clinical Evaluation, Self-Reported History"

_ijms, 2024, doi:10.3390/ijms25031599_

Round 1

Reviewer 1 Report

Comments and Suggestions for Authors

In their paper on a large population on lipedema Patton et al. give a detailed description of the patients with this disease. This description includes phenotype and staging, onset and evolution of the disease, family history, endocrinological and lymphological aspects, thickness of the subcutaneous tissue, diverse blood parameters and inflammation markers, lipid and glucose metabolism; vitamin D, hormones, menstrual cycle alterations and polycystic ovary syndrome, gynecological problems, metabolic syndrome, obesity, allergy, and mood disorders. All these aspects are well described and presented and thus the study, though entirely descriptive, provides loads of putatively valuable data that are worth to be published. These data may finally lead to an improvement of clinical evaluation of lipedema patients by a more systematic and holistic approach, particularly by including the endocrine and metabolic aspects. Addressing these aspects might improve the therapeutic success. Though, however, much research is still warranted on this issues.

Minor comment: Figure 2 should contain error bars or other markers of deviation.

Author Response

In their paper on a large population on lipedema Patton et al. give a detailed description of the patients with this disease. This description includes phenotype and staging, onset and evolution of the disease, family history, endocrinological and lymphological aspects, thickness of the subcutaneous tissue, diverse blood parameters and inflammation markers, lipid and glucose metabolism; vitamin D, hormones, menstrual cycle alterations and polycystic ovary syndrome, gynecological problems, metabolic syndrome, obesity, allergy, and mood disorders. All these aspects are well described and presented and thus the study, though entirely descriptive, provides loads of putatively valuable data that are worth to be published. These data may finally lead to an improvement of clinical evaluation of lipedema patients by a more systematic and holistic approach, particularly by including the endocrine and metabolic aspects. Addressing these aspects might improve the therapeutic success. Though, however, much research is still warranted on this issues.

We thank the reviewer for his/her positive comment, and we agree that it should be only the first step to better understand lipedema

Minor comment: Figure 2 should contain error bars or other markers of deviation.

Probably you mean Fig 5 (fig 2 is a picture), we added deviation bars

Reviewer 2 Report

Comments and Suggestions for Authors

Patton et al. conducted a study investigating lipedema and its associated biochemical and anthropometric changes in a cohort of 360 Italian women aged 12 to 76 years. The research addresses significant gaps in existing data on the subject. However, I have several important concerns about the study:

  1. Many biochemical parameters, including glycemic, hormonal, and inflammatory factors, lose significance when corrected for BMI. This raises doubts about whether these associations are linked to obesity rather than lipedema. Have the authors attempted to adjust these associations for WHR?

  2.  
  3. The authors mentioned evaluating the family history of lipedema among their patients. Were the 360 studied women unrelated, or did the study include mother-daughter or sister-sister pairs?

  4.  
  5. Was physical activity considered in all the comparisons made? Physical activity can play a crucial role in lipid accumulation. Were the participants leading a more sedentary or active lifestyle?

  6.  
  7. Parameters such as WHR, WHTR, waist and hip circumference are predictors of abdominal obesity and may not be ideal for studying thigh lipid accumulation. Have the authors considered using thigh circumference in conjunction with ultrasound measurements?

  8.  
  9. Why were alopecia and acne examined in relation to lipedema? What is the rationale behind exploring these specific conditions?

  10.  
  11. What were the exclusion criteria for the samples? Were pregnant women excluded from the study?

The introduction is very lengthy and repetitive.

Comments on the Quality of English Language

Minor editing needed

Author Response

Patton et al. conducted a study investigating lipedema and its associated biochemical and anthropometric changes in a cohort of 360 Italian women aged 12 to 76 years. The research addresses significant gaps in existing data on the subject. However, I have several important concerns about the study:

We thank the reviewer for his/her constructive comments that will improve the quality of our manuscript.

Many biochemical parameters, including glycemic, hormonal, and inflammatory factors, lose significance when corrected for BMI. This raises doubts about whether these associations are linked to obesity rather than lipedema. Have the authors attempted to adjust these associations for WHR?

We thank the reviewer for the comment; the correction with BMI to point out the worsening via lipedema’s stages linked to obesity; we also evaluated the correction with the WHR, but this does not change the result except for the IGF1; we decided not to include this data because the WHR is even less indicative for lipedema, as the waist and hips are often not affected.

The authors mentioned evaluating the family history of lipedema among their patients. Were the 360 studied women unrelated, or did the study include mother-daughter or sister-sister pairs?

We thank the reviewer for the comment; the sample was almost unrelated, being the familiarity cases are under 5%

Was physical activity considered in all the comparisons made? Physical activity can play a crucial role in lipid accumulation. Were the participants leading a more sedentary or active lifestyle?

We thank the reviewer for the comment; we strongly agree on the importance of physical activity; anyway, in our finding, it was difficult to analyze what kind of exercise was practiced, so instead of a simple yes or no, we preferred to exclude it; we are working on a different paper on physical exercise.

Parameters such as WHR, WHTR, waist, and hip circumference are predictors of abdominal obesity and may not be ideal for studying thigh lipid accumulation. Have the authors considered using thigh circumference in conjunction with ultrasound measurements?

We thank the reviewer for the comment. yes, we did, and we agree that the actual parameters are not fit for lipedema, but at the moment, others circumference or parameters need more validations to be taken in count

Why were alopecia and acne examined in relation to lipedema? What is the rationale behind exploring these specific conditions?

It is a routine in endocrine evaluation, so we reported it as a possible hallmark of a possible hormonal imbalance.

What were the exclusion criteria for the samples? Were pregnant women excluded from the study?

There were no exclusion criteria, being an observational study; no one of the patients was pregnant (last three months before)

The introduction is very lengthy and repetitive.

Thanks. We reduced it and tried to make it more readable.

Round 2

Reviewer 2 Report

Comments and Suggestions for Authors

None